# FineWeb2: One Pipeline to Scale Them All — Adapting Pre-Training Data Processing to Every Language

**Guilherme Penedo**[🤗]* **Hynek Kydlíček**[🤗] **Vinko Sabolčec**[EPFL] **Bettina Messmer**[EPFL]
**Negar Foroutan**[EPFL] **Amir Hossein Kargaran**[🤗] **Colin Raffel**[🤗] **Martin Jaggi**[EPFL]
**Leandro Von Werra**[🤗] **Thomas Wolf**[🤗]
[🤗]Hugging Face  [EPFL]EPFL

[⌂] Pipeline code: https://github.com/huggingface/fineweb-2
[🤗] FineWeb2 dataset: https://hf.co/datasets/HuggingFaceFW/fineweb-2

## Abstract

Pre-training state-of-the-art large language models (LLMs) requires vast amounts of clean and diverse text data. While the open development of large high-quality English pre-training datasets has seen substantial recent progress, training performant multilingual LLMs remains a challenge, in large part due to the inherent difficulty of tailoring filtering and deduplication pipelines to a large number of languages. In this work, we introduce a new pre-training dataset curation pipeline based on FineWeb (Penedo et al., 2024) that can be automatically adapted to support any language. We extensively ablate our pipeline design choices on a set of nine diverse languages, guided by a set of meaningful and informative evaluation tasks that were chosen through a novel selection process based on measurable criteria. Ultimately, we show that our pipeline can be used to create non-English corpora that produce more performant models than prior datasets. We additionally introduce a straightforward and principled approach to rebalance datasets that takes into consideration both duplication count and quality, providing an additional performance uplift. Finally, we scale our pipeline to over 1000 languages using almost 100 Common Crawl snapshots to produce FineWeb2, a new 20 terabyte (5 billion document) multilingual dataset which we release along with our pipeline, training, and evaluation codebases.

## 1 Introduction

One of the main drivers of the improving capabilities of large language models (LLMs) is increased scale, in terms of both model and pre-training dataset size. To satiate the ever-growing hunger for text data, most pre-training datasets include large amounts of text scraped from the public internet (Raffel et al., 2020; Penedo et al., 2023; 2024). Consequently, pre-training data tends to be most readily available in the "high-resource" languages (English, Chinese, etc.) that are most prevalent on the internet. Since LLM capabilities largely stem from the data they were trained on (Grosse et al., 2023; Roberts

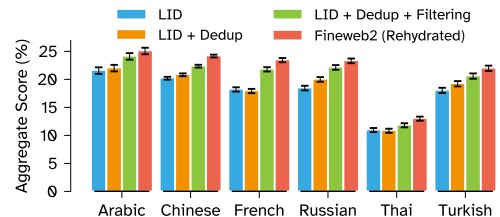

Figure 1: **The FineWeb2 pipeline:** Evaluation results of models trained on 350 billion tokens show that each pipeline step – Language Identification (**LID**), Deduplication (**Dedup**), **Filtering**, and Dedup-informed upsampling (**Rehydration**) – improves performance.

et al., 2020; Razeghi et al., 2022), this has resulted in language models having better performance on high-resource languages. Furthermore, commercial and open language model development frequently only targets these languages (Grattafiori et al., 2024; Jiang et al., 2024; 01.AI et al., 2025). This state of affairs leaves the majority of the world's population

---

*Correspondence to guilherme at huggingface dot co

(speaking over 7,000 languages (Eberhard et al., 2024)) unable to interact with state-of-the-art LLMs in their native tongue.

Why not just curate datasets in underrepresented languages and train LLMs on them? Putting aside the possible lack of data (recent LLM training runs typically require trillions of tokens (AI@Meta, 2024; DeepSeek-AI et al., 2024)), a key challenge is that high-resource languages benefit from the existence of well-tuned and battle-tested data processing and curation pipelines, whereas low-resource languages face a vastly different landscape: evaluating corpora quality, ensuring accurate language identification, customizing filtering recipes, and even separating words can be major challenges for many languages. While some past work has successfully curated single-language pre-training datasets and used them to produce strong language-specific models (Carmo et al., 2020; de Vries et al., 2019; Le et al., 2020; Martin et al., 2020; Delobelle et al., 2020; Luukkonen et al., 2023; PLLuM Consortium, 2025; Pipatanakul et al., 2023, etc.), hand-designing a different pipeline for each language does not scale. Consequently, most past work on multilingual datasets (e.g. Xue et al., 2021; Wenzek et al., 2020; De Gibert et al., 2024) has used a (mostly) fixed pipeline across all languages. This one-size-fits-all approach risks applying inappropriate filtering to different languages, obviating the goal of creating performant data in many languages.

In this work, we introduce a new data processing pipeline based on the approach used for the state-of-the-art English pre-training dataset FineWeb (Penedo et al., 2024). Importantly, our pipeline can be *automatically adapted* based on language-specific statistics to produce high-quality pre-training corpora in any language. We follow a data-driven approach and validate our design choices by running extensive ablation experiments where we train monolingual models on a set of nine diverse languages and evaluate on tasks chosen through a novel selection process based on measurable criteria that ensure a meaningful signal. In addition, we introduce a straightforward and principled approach to rebalance datasets using the original duplication counts and quality signals that allows globally near-deduplicated datasets to obtain a performance uplift. Ultimately, we show that models trained on language-specific corpora produced by our pipeline perform better than those trained on other public web-based multilingual datasets by training models on additional "unseen" languages that were not used to inform pipeline design decisions. Finally, we use our pipeline to process almost 100 Common Crawl[1] snapshots spanning the summer of 2013 to April 2024 to create FineWeb2, a new 20 terabyte (5 billion document) dataset covering over 1000 languages. FineWeb2 is released under the permissive ODC-By License, and we additionally release the pipeline, training, and evaluation codebases, as well as the preliminary version of the dataset obtained after the deduplication stage, to facilitate further research on multilingual pre-training dataset curation.

## 2 Preliminaries

Before detailing our dataset creation process, we first establish critical considerations that arise when dealing with massively multilingual data.

**Notation** When considering thousands of languages, it's important to have an unambiguous way of referring to languages and scripts. In our work we identify languages by their official ISO-639-3 codes[2] which cover significantly more languages than the commonly used ISO-639-1 codes (such as "en", "zh", etc). As many languages use multiple writing systems (scripts), we optionally designate individual "languages" by a *(ISO-639-3 language code, ISO 15924 script code)* pair. For instance, 'arb_Arab' is Standard Arabic in Arabic script, while 'arb_Latn' is Standard Arabic in Latin script.

**Separating words** Many parts of our processing and evaluation pipeline require the ability to separate (tokenize) text into individual words. For example, we rely on word tokenization when we filter documents based on the ratio of words that have a given property, when selecting n-grams for deduplication, or even when evaluating generative tasks. While

---

[1] https://www.commoncrawl.org/
[2] https://iso639-3.sil.org/code_tables/639/data

whitespace and punctuation often mark word boundaries, many writing systems use different boundary markers or have no visible markers at all (Daniels & Bright, 1996). This is particularly common in Southeast Asian languages, as well as Chinese, Japanese, and Korean. Therefore, word tokenizers/segmentators tailored to each language and script are needed. We collected a large number of tokenizers from SpaCy (Honnibal et al., 2020) and Stanza (Qi et al., 2020), as well as from libraries targeting specific languages (or language groups). We then assigned proxy tokenizers based on the closest language according to language family data from the Ethnologue[3] to languages without a native word tokenizer. For more details on this process, see Appendix A.1. These tokenizer assignments were crucial to adapt filtering, deduplication, and evaluation setups to thousands of languages.

## 3  Experimental setup

To compare and validate pipeline design choices, we followed an experimental setup similar to Penedo et al. (2024). Specifically, to assess data quality, we relied on training small models and evaluating them on "early-signal" benchmark tasks, i.e., tasks where models perform reasonably well after only a few tens of billions or hundreds of billions of training tokens, allowing us to confidently establish comparisons between them. For each processing step, we conducted comparative evaluations using two identical models that differed only in their training data: one model was trained on data with the processing step applied, while the other used the unprocessed (ablated) version. By keeping all other variables constant (number of parameters, architecture, tokenizer, and training token count), we could isolate the impact of each data processing step on downstream model quality.

While ideally we would have tested each processing step across every language, computational constraints and the lack of evaluation tasks for many of the languages made this impractical. We therefore chose to conduct our experiments on a select set of nine **canary languages** (i.e. test languages): Arabic, Chinese, French, Hindi, Russian, Swahili, Telugu, Thai, and Turkish. Testing across these languages allowed us to evaluate the impact of each design decision across different language families, scripts, and levels of resource availability, while keeping computational requirements manageable. These details are available on Table 1, where Resource Availability was determined following Joshi et al. (2020). We trained separate models per language, rather than a single multilingual model, to avoid introducing confounders between languages. This means that for every ablation experiment or validation run reported in this paper, *we trained nine different models (one per language)*.

### 3.1  Tokenizer and model architecture

**Tokenizer**    The choice of tokenizer can induce differential downstream model performance across different languages based on how compactly it maps a given language's words into tokens (Mielke et al., 2021). Given that our experiments target different languages and, in particular, different scripts, we evaluated the **subword fertility** and **proportion of continued words** (Rust et al., 2021) of different existing open-source tokenizers from leading multilingual LLMs on our nine canary languages. Concretely, we split text from each language's Wikipedia into individual "real" words using our word-level tokenizers (discussed in Appendix A.1) and then measured the average number of tokens per word for each tokenizer. From the tokenizers that showed reasonable fertility on our nine canary languages, we chose the tokenizer used in **Gemma** (Gemma Team et al., 2024), a modern tokenizer with a vocabulary size of around 250,000 tokens that showed better average fertility than similarly sized tokenizers. Detailed results are available in Appendix A.3.

**Model architecture**    We used a similar model architecture setup to Penedo et al. (2024), with a reduced number of layers given the additional embedding parameters due to the larger vocabulary size. All models used in our experiments were trained using the nanotron training framework, and followed the Llama (Touvron et al., 2023) architecture with 14 layers,

---

[3]https://www.ethnologue.com/browse/families/

32 attention heads, length-2048 sequences, and tied embeddings, for a total of **1.46 billion** parameters. Further details and training hyperparameters are provided in Appendix A.4.

## 3.2 Baseline datasets

We selected existing widely used multilingual datasets to use as comparison baselines. For each language, we trained one model on language-specific data from each reference dataset: **CC-100** (Wenzek et al., 2020; Conneau et al., 2020), **mC4** (Xue et al., 2021), **CulturaX** (Nguyen et al., 2024), and **HPLT** (de Gibert et al., 2024). We additionally trained multiple models on "raw" Common Crawl data (after text extraction and Language Identification, but without any additional filtering or deduplication). Unfortunately, all datasets except raw Common Crawl only contained a limited amount of data for Telugu and Swahili, and only CulturaX and HPLT had enough data for a pre-training run in Hindi at 30 billion tokens without requiring an excessive number of epochs over the training data.

## 3.3 Selecting evaluation (Fine)tasks

The selection of English evaluation tasks is straightforward due to the existence of well-established benchmarks such as MMLU (Hendrycks et al., 2021) or HellaSwag (Zellers et al., 2019), which are widely used and supported by all major evaluation frameworks. The situation is significantly different for non-English languages, which often lack evaluation tasks. When available, these tasks often lack broader community validation and suffer from quality issues – many are machine-translated and may even include English words in their formulations (Artetxe et al., 2020b). Additionally, we find that non-English tasks are often unsuitable for early pre-training evaluation due to suboptimal task formulations and/or excessive difficulty that results in random-level performance.

To identify informative evaluation tasks, we established four key criteria for what we call **early-signal** tasks: **Monotonicity** – the performance of models evaluated on this task should improve as training progresses, though possibly at different rates depending on the pre-training dataset; **Low noise** – when comparing models trained on different datasets, we want to ensure that the relative performance differences between them are due to inherently better training data, and not due to evaluation noise; **Non-random performance early in training** – tasks reflecting model capabilities that are only acquired later in training are not informative for small scale pre-training ablations, as near-random scores cannot meaningfully differentiate between datasets; **Ordering consistency** – if model A outperforms model B, then falls behind, then leads again within a short span of training steps, we cannot confidently determine which model (and, correspondingly, dataset variant) is superior and we therefore need tasks that provide consistent relative performance.

We defined quantitative metrics to measure these characteristics and applied them to hundreds of candidate zero-shot evaluation tasks targeting our 9 canary languages on the models trained on our baseline datasets. See Appendix A.5 for the precise definition of "early-signal" tasks and additional description of our evaluation setup. We strove to cover **different task types** in all languages: Reading Comprehension, **RC**; General Knowledge, **GK**; Natural Language Understanding, **NLU**; and Common-Sense Reasoning, **CR**.

Our in-depth analysis of existing evaluation tasks resulted in a final suite of **84** selected benchmarks out of **197** tested across our nine canary languages. We list all the tasks and employed metrics in Appendix A.5.3.

To produce an **aggregate score** across tasks, we follow the approach used by Fourrier et al. (2024); Li et al. (2024b) and average scores across tasks after first rescaling scores based on the random baseline – any score below the random baseline is considered 0, and for the remaining scores we subtract the random baseline value and shift the scores as $new\_score = (score - random\_baseline)/(1 - random\_baseline)$. As some languages might have an unbalanced number of tasks for each task category (RC, GK, NLU and CR), during score averaging we first average within categories themselves and then take the average of each category. This per-category macro-average score is our final reported aggregate score.

# 4  The FineWeb2 pipeline

## 4.1  Starting point: FineWeb

We started by applying the first few processing steps used in the creation of the English-only FineWeb dataset (Penedo et al., 2024): downloaded WARC (web archive) files from all available (almost 100) CommonCrawl snapshots, applied URL filtering using a blocklist to remove adult content (an approach discussed in Penedo et al. (2023)), and used trafilatura (Barbaresi, 2021) to extract text content from the HTML in the WARC files. We then aimed to adapt the remaining components of the FineWeb pipeline – filtering and deduplication – starting with all the data that was excluded during FineWeb's language filtering step (which uses the FastText language identifier (Joulin et al., 2016) to identify English text with a threshold of 0.65). Since approximately 40% of all documents met the FineWeb English language threshold, our starting point for FineWeb2 comprises the remaining 60% of all the text extracted from CommonCrawl content.

## 4.2  Language Identification (LID)

A critical first step for curating a multilingual dataset from web scrapes is accurately identifying the main language of each document. The choice of Language Identification (LID) tool determines not only how reliably each language (label) is predicted, but also the set of identifiable languages – if the LID does not have a label for a specific language, then its content will either be removed or misclassified as some other language. Additionally, as LID classifiers usually assign a confidence score to each prediction, the choice of filtering thresholds further affects the amount of data retained, as well as its quality, as LID confidence can often be correlated with the noisiness of a given document (NLLB Team et al., 2022).

**Choice of classifier**   While Transformer-based LID classifiers exist (Bapna et al., 2022), they are too slow and expensive to run at a large scale. Most commonly used LID classifiers are simple character level n-gram models, including CLD3 (Salcianu et al., 2018) (107 supported languages, used in mC4 (Xue et al., 2021)) and classifiers following the fastText architecture (Joulin et al., 2016), such as FT176[4] (176 languages, used in CC-100 (Wenzek et al., 2020; Conneau et al., 2020) and CulturaX (Nguyen et al., 2024)), OpenLID (Burchell et al., 2023) (193 languages, used in HPLT2 (Burchell et al., 2025)), and the recent GlotLID (Kargaran et al., 2023) (1880 languages). Although FineWeb Penedo et al. (2024) used FT176, using GlotLID would allow us to support a much larger number of languages, as well as to run separate processing for different scripts of the same language, as GlotLID explicitly separates them. Additionally, it includes special labels for non supported scripts and for common formats of "noise" documents, preventing this content from being classified as one of the other languages.

While GlotLID reports strong performance on language classification benchmarks and supports a large number of languages, we are primarily interested in the downstream model quality resulting from using a given LID tool. Therefore, for each canary language we trained one model on documents classified as this language (regardless of confidence) by FT176 and another based on GlotLID. We then evaluated the models on our set of evaluation tasks and found that GlotLID outperforms FT176 (Fig. 5) on higher resource languages while being slightly behind on lower resource languages. We consider the increased language coverage to make up for this difference and employ GlotLID for our pipeline. See Appendix A.6.1 for additional discussion and results.

**Confidence thresholds**   In addition to providing the most likely language of a document, LID classifiers typically also return a confidence threshold for that prediction. Many works rely on a single confidence threshold applied to all languages, e.g., in mC4 (Xue et al., 2021) only documents whose language prediction score is above 70% are kept, while in CC-100 (Wenzek et al., 2020) a score of 50% is used for all languages. However, this does not account for inherent differences in prediction confidence between languages – some

---

[4]https://fasttext.cc/docs/en/language-identification.html

languages have a closely related cousin that might confound the LID classifier, therefore requiring a lower threshold, whereas a higher value can be employed for high resource languages for which the classifier is often quite confident (NLLB Team et al., 2022). To determine appropriate thresholds per language following our data-driven philosophy, we train models for each of our nine languages at different confidence thresholds, corresponding to removal rates of 5% of the data at a time.

Languages such as Arabic (Table 16) or Russian (Table 20) prefer high thresholds (>0.8), while for Swahili a lower threshold around 0.3 (corresponding to a removal rate of almost 65%) performs best, as this language's distribution is right-skewed. After analyzing the score distributions and the highest performing thresholds, we defined filtering thresholds to be one standard deviation below the median of the score distributions, clipped to the range $[0.3, 0.9]$: $\max\{0.3, \min\{0.9, \text{Med}(X) - \sigma(X)\}\}$, where $X$ is the distribution of confidence scores for this language's data. We found that this formula selects values within the highest performing threshold regions for most languages (Table 15).

### 4.3 Deduplication

Deduplication is the process of removing highly similar documents from a pre-training dataset to increase training efficiency and improve model performance (Lee et al., 2022). While deduplication requires a large amount of computation and is therefore typically applied as the very last processing step, we employ it as an initial step, before filtering. This allowed us to directly observe the final dataset performance each time we ran one of our many filtering experiments without the possibility of deduplication later influencing the results.

We rely on MinHash (Broder, 1997), a "fuzzy" deduplication method that finds clusters of similar documents that are then filtered to keep a single document per cluster. We used the same MinHash hyperparameters used for FineWeb (14 buckets of size 8, with 5-grams) and deduplicated globally per language. We used our word-level tokenizers (Section 2) to obtain word n-grams. When keeping a single document per duplicate cluster, we record the number of documents that were in the cluster to explore duplication-aware upsampling schemes later in Section 4.5.

To measure the impact of deduplication on data quality, we trained per-canary-language models on 350 billion tokens, both on the data before deduplication (with the LID filtering) and after. Results in Fig. 1 show that while we generally observed improved performance across languages, the impact of deduplication seems to vary significantly from language to language, without any discernable relationship to the language's resource level. However, we note that even languages showing little to no improvement from deduplication still benefit from rehydration (our duplication-aware upsampling scheme, described in Section 4.5).

### 4.4 Filtering recipe

Filtering aims to remove documents that are deemed to be "lower-quality" (i.e. those that might worsen model performance) using heuristic rules, such as the number of times words are repeated within the document, the average number of characters per words in the document, or the ratio of lines ending with punctuation Albalak et al. (2024). Unfortunately, many of these rules are language-specific: in languages like Chinese, words have, on average, fewer characters, while in languages like German the opposite is true.

We began with the list of filtering rules from FineWeb and sought to devise methods that would allow us to automatically adapt them to a large number of languages, tailoring specific thresholds according to each language's characteristics. To this end, we collected statistics for each language on different corpora and used the distributions on different metrics to determine adequate filtering thresholds. We relied on our nine canary languages to inform our decisions and trained a large number of models to test how well each rule adaptation method would generalize. We leveraged three main sources to collect statistics for each language: Wikipedia, the Glotlid-Corpus (Kargaran et al., 2023) (used to train the GlotLID classifier) and our language-filtered data obtained from Common Crawl.

### 4.4.1 Stopwords

Stopwords are common words in a language that, while not indicative of text quality, when absent can help identify non-linguistic "low-quality" content (e.g. boilerplate, non-natural text, or gibberish), or content whose language was misclassified. The number of stop words in a document is therefore used as a signal to remove such data, and stopword filtering is part of the widely used Gopher quality filters (Rae et al., 2022) for English.

To determine stopwords for each language, we analyzed word frequencies in our reference datasets, using our word tokenizers to identify the most frequently occurring words. Instead of selecting a fixed number of words, we defined stop words as those exceeding a set frequency threshold. This method allowed us to account for variations across languages. For example, in English, "the" is highly frequent, whereas in German, its equivalents—"der," "die," and "das"—share the same role. We additionally addressed specific issues: some "words" were actually non-alphabetic and had to be excluded, and for some languages the source data (particularly Wikipedia) contained large portions of English content that caused a significant number of the stop words to be in English. This underscores the importance of having clean data when creating filters in an automated fashion. Further discussion in Appendix A.7.1. For our filtering pipeline, we require at least 2 words from the stopwords list to be present in each document, in line with Rae et al. (2022).

### 4.4.2 Filtering threshold selection

To automatically determine filter thresholds for different languages, we propose an empirical approach based on the distribution of the metric we are filtering. We consider a variety of different methods: **English**, use English-based filtering values from FineWeb without change (one of the baselines); **MeanStd**, assuming the threshold is $n$ standard deviations from the mean in the metric distribution in English, we set the threshold to the corresponding distance from the mean in the target language distribution (a variation using the median instead of mean produces similar values); **Quantile**, where we define the threshold for each language so as to remove the same fraction of data as the English threshold removes in English; **10Tail**, inspired by CulturaX (Nguyen et al., 2024), we select a threshold to remove the 'tail' – exactly 10% – of the reference data; **MedianRatio**, inspired by HPLT2 (de Gibert et al., 2024), thresholds are selected such that the ratio between English and the target language matches the ratio of the medians of English and the target language on this metric. For each method, thresholds are computed on different reference corpora for each filter and then models are trained on the data filtered using these filters. We then compare method for each filter across all languages with each other, as well as with a "no filtering" baseline.

Precisely, we computed thresholds for each filter used in three of the FineWeb filter groups: Gopher Quality (goq), Gopher Repetition (gor), and FineWeb Quality (fwq). We then trained nine models (one per canary language) on data filtered using each method on each of the filter groups, for all method-filter group combinations except those that removed an excessive amount of data (more than 75%), or that did not remove any data at all. In total, these experiments required a total of 207 ablation models, each trained for 29B tokens. We report the average rank of the aggregate score of each method across languages, in Table 25. Ultimately, we employ the best performing methods for each filter group: the **10Tail** method and **Quantile** methods computed on Wikipedia (or on GlotLID-Corpus for languages without a Wikipedia) for the FineWeb and Gopher Quality filters, respectively, and the **MeanStd** method computed on Common Crawl data for the Gopher Repetition filters. This step noticeably improves performance for all languages (Fig. 1).

### 4.4.3 Precision filtering lower-resource languages

Low-resource languages often suffer from low LID precision: due to the large class imbalance between high- and low-resource languages on web corpora, real precision is often much lower than that measured on a balanced test set (Caswell et al., 2020). In practice, this means that corpora for low-resource languages with a closely related high-resource language are often heavily contaminated with false positives from the high-resource language, sometimes accounting for more than 90% of the data.

After inspecting data for low-resource languages produced by our pipeline, we decided to employ a final filtering step exclusively to low-resource languages to address this issue. Inspired by Caswell et al. (2020); Bapna et al. (2022), we compiled lists of words that are common in each language but uncommon in other languages (i.e., have high affinity). We then measured the "contamination" of each corpora as the ratio of documents not containing any of these words. While the majority of languages had extremely low contamination scores, roughly a third of the 1900 languages had contamination scores above 10%. For these languages, we filtered documents using the high-affinity wordlists to remove false positive documents. Additionally, since we noticed the high-affinity wordlists could be too short and strict for some languages (such as English-based pidgins, for example), we also kept documents removed by the wordlist filtering whose URLs included specific terms related to the language (the language code, the language name, domain name extensions etc). A manual audit of three lower-resouce languages shows precision improvements of almost 30% for some languages. We provide additional details in Appendix A.7.3.

### 4.5 Rehydration

In contrast to standard deduplication practices (Lee et al., 2022), Penedo et al. (2024) makes the case for per-snapshot deduplication and claims that additional deduplication beyond the removal of the largest duplicate clusters may actually harm model performance by artificially upsampling documents that are completely unique but high-entropy and low-quality. While we perform global deduplication, as mentioned in Section 4.3, we also save the original size of each duplicate cluster in the metadata of the kept documents, which allows us to selectively upsample specific documents (and therefore "rehydrate" the dataset), to obtain more performant models.

In Tang et al. (2024), the authors explore one such strategy with hand-picked upsampling weights based on MinHash cluster sizes: documents with 2 to 5 duplicates are repeated 3 times, 5-100 5 times, 101-1000 8 times, and documents with over 1000 duplicates are repeated 10 times. While this provides a duplication-aware upsampling strategy, it is heavily dataset-dependent – smaller datasets will have their distribution of cluster sizes shifted left – and therefore might not be scalable across different languages. Additionally, the chosen weights favor highly duplicated documents the most, which we find are generally of lower quality, and therefore should be repeated less rather than more.

While we initially trained models for each of our nine canary languages on data of different ranges of minhash cluster sizes (e.g., we trained one model on data that had no duplicates, another on data that had 2 duplicates, data that had 3-4 duplicates, etc) to empirically define upsampling weights, a simpler and more scalable approach is to use the results from our filtering stage as a proxy for cluster size quality: we obtain the global filtering rate (the percentage of documents removed by our entire filtering process), as well as the filtering rate for each value of metadata minhash cluster size, as shown in Fig. 2 (for French).

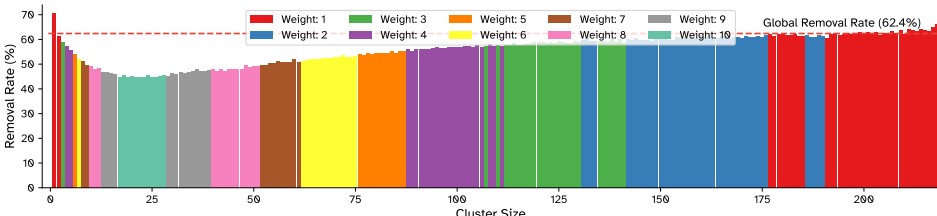

Figure 2: **Filtering rates by MinHash cluster size** for French documents. The global filtering rate represents the overall percentage of documents removed during the full filtering process. Individual filtering rates are shown for each cluster size, providing a proxy for cluster quality—higher removal rates may indicate lower-quality clusters. We assign **upsampling weights** to each cluster size based on the filtering rates.

The figure suggests that both data that was never repeated (cluster size of 1), as well as data that is repeated many times (especially the most-repeated 0.1% "long tail" of data grouped

as the last bar), is generally of lower quality, as our filters removed more than the global removal rate. Surprisingly, this U looking shape we observe for French is present in most languages we verified, but often shifted based on the size of the corpora for each language.

The differences we observe for different cluster sizes align closely with experimental results from training runs on different ranges of cluster sizes for the languages we tested, and so we experimented with setting upsampling weights based on the removal rates: we assigned a weight of 10 (meaning documents should be repeated 10 times) to the cluster size with the smallest removal rate, and a weight of 1 to every cluster size above the global removal rate. For the remaining cluster sizes, we resorted to simple interpolation between these 2 endpoints. For French, the resulting weights are shown in Fig. 2. While upsampling weights are dataset-dependent, using the filtering rates as a proxy for quality is a scalable and affordable method to determine them and rehydration itself generally provides a strong performance uplift (Fig. 1) with little downside.

## 5 Validating and Applying the FineWeb2 Pipeline

Having established the pipeline for FineWeb2, and having shown the positive effect of each pipeline step (Fig. 1), we now perform additional evaluations to confirm the effectiveness of our approach and use the pipeline to generate per-language datasets in over 1,000 languages.

**Creating the FineWeb2 dataset**   We apply our pipeline to 96 Common Crawl snapshots, spanning the summer of 2013 to April 2024, to produce the FineWeb2 dataset, comprising 20 terabytes of text content covering a total of 1,868 language-script pairs, of which 1,226 have over 100 documents, 474 more than 1 thousand documents, and 203 at least 10 thousand documents. Additional details and per-language statistics can be found in Appendix A.11. In addition to the filtered dataset, we also release the preliminary version before filtering is applied, to facilitate further research into alternative filtering methods. As FineWeb2 itself does not include English, for full language coverage we recommend complementing it with FineWeb, whose pipeline inspired FineWeb2.

**Comparison to other datasets**   We now compare to other non-English datasets, both on the canary languages used to design the pipeline as well as a set of unseen languages that were not used for ablations. As discussed previously, prior multilingual datasets often use fixed pipelines across languages, whereas FineWeb2's pipeline adapts to the statistics and characteristics of each language. By comparing to other multilingual datasets, we can confirm the benefit of FineWeb2's adaptive approach. To provide a point of comparison against pipelines tuned to a specific language, we additionally evaluate single-language datasets (whose pipelines are designed and tuned for a specific language, often by native speakers) when available. For canary languages, we use the same set of benchmarks used for pipeline design ablations. Since the FineWeb2 pipeline was designed specifically around the canary languages, evaluating on unseen languages validates that the pipeline generalizes effectively. To choose unseen languages, we first followed the same procedure (detailed in Section 3.3) for selecting reliable evaluation tasks across a wide range of languages and chose languages that had a sufficient number of reliable tasks: German, Indonesian, Italian, Japanese and Vietnamese. The chosen tasks are detailed in Appendix A.10. Canary-language and unseen-language models were trained for 29 billion and 100 billion tokens respectively. All evaluated models follow the same architecture, hyperparameters, and (Gemma) tokenizer as considered previously and detailed in Section 3.1.

A summary of the results is shown in Fig. 3, with detailed per-task results in Appendix A.10.2. Overall, we found that FineWeb2 produces more performant models than prior multilingal datasets on 11 out of 14 of the languages we considered. In some cases, FineWeb2 produces worse performance than a language-specific dataset, which highlights that pipelines hand-designed by language experts can still outperform our adaptive pipeline approach. These trends hold up both for our canary datasets as well as held-out datasets, which supports the utility of the 1,000+ language-specific datasets we generated with the FineWeb2 pipeline. On the whole, our results confirm the effectiveness and generalization of our consistent-but-adaptable cross-lingual curation pipeline.

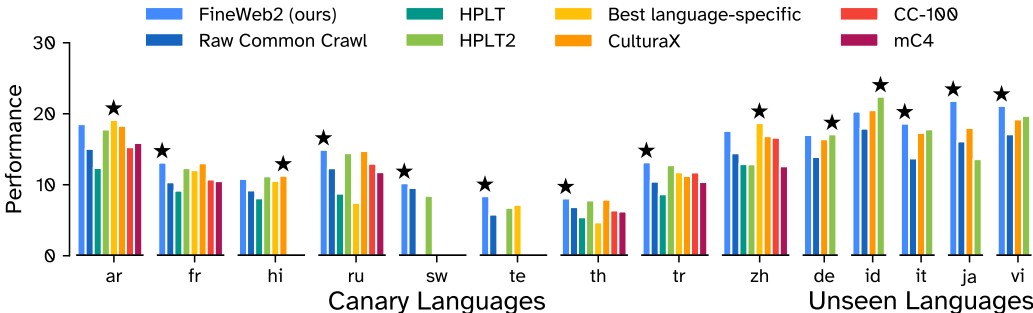

Figure 3: High-level performance comparison of FineWeb2 to other multilingual and language-specific datasets. We evaluate performance both on the canary languages used to design the FineWeb2 pipeline as well as unseen languages. For brevity, for each language we plot the performance of only the best-performing single-language dataset. The best-performing dataset for each language is marked with ★. Expanded results are provided in Appendix A.9 and Appendix A.10.2.

**Inspecting low-resource corpora**   A natural concern is whether low-resource corpora, often with fewer than 20 documents, contain content that is genuinely useful for training. Manual inspection of over 500 languages reveals that many corpora are composed almost exclusively of Bible and/or Wikipedia content. We categorized the most common document domain names and computed the proportion belonging to Bible- or Wikipedia-related sources: out of 1868 language-script pairs in the final dataset, 70% (1320 of them) have more than half their documents from Bible- or Wikipedia-related domains. This reflects both the limited availability of online data for many languages and the narrow diversity of sources in the language identifier's training data—where often the only "clean" data comes from the Bible (Kargaran et al., 2023). While we hope these corpora remain useful to the research community, their limited diversity highlights the broader challenges of collecting data for the long tail of the world's languages. For more details, see Appendix A.12.

## 6   Conclusion

In this paper, we used a data-driven approach to design a multilingual pre-training data processing pipeline that can automatically adapt to all languages, in contrast to prior work that employs fixed pipelines for each language. We extensively ablate our design choices on a new suite of quantitatively identified multilingual benchmarks that provide a reliable evaluation signal, ultimately covering 14 languages. We additionally show how duplication counts and filtering results can be leveraged to selectively upsample higher quality content, providing a performance uplift. Finally, we scaled our pipeline to create FineWeb2, a pre-training dataset covering 1,868 language-script pairs, spanning 20 terabytes of text content curated from 96 Common Crawl snapshots.

While our experiments show that our pipeline yields strong performance, we point out a few limitations. First, although we strove to make the language coverage as wide as possible, computational constraints, language-specific task availability, and excessively small low-resource datasets only enabled us to test a small proportion of the languages in FineWeb2. These factors also forced us to only consider relatively short ablation runs. Second, we studied "early-signal" properties of each task at the very early stages of model training, and so it is possible that the properties could change significantly as training progresses, making some tasks more viable. Additionally, we do not explore additional criteria for task selection, such as "cultural alignment", with which translated tasks struggle. Similarly, our chosen tasks do not measure other important attributes such as bias or diversity. Lastly, while we strove to include a large number of low-resource languages in our dataset, a large number of them consist almost or even entirely of Bible- or Wikipedia-related content. Overall, we hope our findings, datasets, and code pave the way for further improvement of datasets that cover a wider range of languages.

## Ethics statement

While the Fineweb2 pipeline incorporates best efforts to support inclusivity and protect personally identifiable information (PII) data, the vast scale of Common Crawl and automatic processing cannot provide guarantees. For this reason, the processing pipeline and datasets—including the removed samples—have been made public, allowing other parties to investigate and improve upon them. Nevertheless, we address risks that could potentially arise from using Fineweb2 more specifically below.

**Bias** Large Language Models are known to reproduce the bias present in their pre-training data (Bender et al., 2021; Feng et al., 2023). As noted by Bender et al. (2021), even petabyte-scale resources such as Common Crawl fail to capture the full range of human perspectives due to disparities in internet participation, leading to the over-representation of certain viewpoints. While our approach does not directly address disparities in internet participation, we curate pre-training data that includes low-resource and historically excluded languages. By broadening the linguistic and cultural coverage, we aim to incorporate diverse viewpoints and support greater inclusivity.

**PII** Subramani et al. (2023) demonstrated that web data contains various types of personal information, including phone numbers, email addresses, IP addresses, and credit card numbers. Research has shown that malicious users can extract sensitive information from LLMs' training corpora by exploiting the models (Carlini et al., 2021; Chen et al., 2023). However, processing Common Crawl at petabyte scale prohibits the use of computationally intensive models, limiting the computational resources that can be used for detecting personal information. Therefore, the FineWeb2 data processing pipeline implements best-effort PII protection using regular expressions, targeting email and IP addresses. More computationally intensive detection methods, the prevention of memorization, or other techniques for responsible LLM training are left to the dataset's end users.

**Toxicity** A challenge in detoxifying language models is that toxicity filters based on word lists of automated methods often incorrectly flag neutral or positive content that mentions marginalized groups, reducing the models' ability to generate text about these groups, even in positive contexts (Welbl et al., 2021). Yet fair treatment of all identity groups is important. Our automated pipeline is designed to work across more than 1000 languages and can be enhanced by the community. We use a URL blocklist to filter unwanted content. This approach helps maintain representation of marginalized groups, while still reducing harmful content in the data curation pipeline.

**Intellectual property** The vast majority of text on the internet is, under most legal frameworks, protected by copyright. The legal implications of training language models on copyrighted text are currently being deliberated in various lawsuits by rights holders against LLM developers (Chat GPT Is Eating the World; Zakrzewski et al., 2024) and consequent risks are therefore unclear (Bengio et al., 2025). Apart from legal risks, content creators have objected to the use of their data on ethical grounds due to the lack of direct compensation for using their content (Baack et al., 2025; Longpre et al., 2024). While Fineweb2 follows current standard practice in not directly addressing these potential risks, the large scale of our data could support efforts to study these issues in more detail (Kandpal et al., 2025).

## Acknowledgments

We would like to thank Abdeljalil El Majjodi, Ihssane Nedjaoui, and Zaid Chiech for labeling data for our precision filtering audit; Bram Vanroy, Loïck Bourdois, Omar Kamali, Per Kummervold, Qian Liu, Edwin Rijgersberg, Michael S. Mollel, Faton Rekathati, and Mikhail Tikhomirov for inspecting and providing valuable feedback on their respective native language subsets of FineWeb2; and the many contributors of the FineWeb-C community annotation project.

We extend our gratitude to the Common Crawl project for freely providing and maintaining their regular crawls, which have enabled much of modern LLM research. We thank Pedro Ortiz Suarez from the Common Crawl team, as well as Gema Ramírez, Marta Bañón, and other members of the HPLT team for fruitful discussions about multilingual data.

Additionally, we thank our colleagues – Nouamane Tazi, Phuc Nguyen, Ferdinand Mom, and Haojun Zhao for designing and building our training framework, Nanotron; Clémentine Fourrier and Nathan Habib for creating and maintaining our evaluation framework, LightEval; and Loubna Ben Allal and Anton Lozhkov for discussions throughout the project. Finally, we thank Hugo Larcher and Mathieu Morlon for tirelessly assisting us whenever we encountered issues with the Hugging Face Science cluster, which they manage with incredible dedication, as well as all the other cluster users for their gracious patience.

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

# A    Appendix

## A.1    Word tokenizers for segmentation

For full language coverage, we relied on a wide range of word tokenizers from SpaCy (Honnibal et al., 2020) and Stanza (Qi et al., 2020), as well as from libraries targeting specific languages (or language groups) like IndicNLP (Kunchukuttan, 2020) for Indic Languages, PyThaiNLP (Phatthiyaphaibun et al., 2024) for Thai, Kiwipiepy (Lee, 2024) for Korean, KhmerNLTK (Hoang, 2020) for Khmer, LaoNLP (Phatthiyaphaibun, 2022) for Lao, Botok (OpenPecha, 2025) for Tibetan, and Pyidaungsu (Sakana, 2024) for Burmese. For languages without a native word tokenizer, we assigned a proxy tokenizer from another language based on language family data from the Ethnologue[5] using the following approach:

1. Build a tree for each language family based on the taxonomy from the Ethnologue.

2. Assign tokenizers to each language+script pair that had a native tokenizer from one of the libraries mentioned above

3. Perform an upward pass through the tree, propagating the available tokenizers to the parent nodes, one per script. When multiple tokenizers for the same script are present in the children of a given node, we pick the one from the language with more available data. We do not propagate to the root node as different subfamilies are usually quite different (for example, for Pidgin, "English-based" and "Swahili-based" are two branches; for Indo-European, "Italic" and "Armenian")

4. Perform a downward pass through the tree, assigning as a proxy tokenizer the previously propagated parent node tokenizers when available.

This method allowed us to quickly scale tokenizer assignments for many languages by assigning tokenizers from a closely related language. An illustrative example is available in Fig. 4. We relied on the SpaCy multilingual tokenizer for the remaining languages with Latin or Cyrillic scripts, which was trained on multiple languages that use these scripts. For any remaining script, we assigned the remaining languages to the tokenizer of the highest resource language that uses the script and has a native tokenizer.

---

[5]https://www.ethnologue.com/browse/families/

## Indo-European Language Family Tree

Figure 4: Example tokenizer assignments based on language family data in Indo-European. Triangles correspond to languages for which a native word tokenizer was available, while squares are languages for which a proxy tokenizer was assigned. The tokenizer assigned to each language is written inside brackets [], and the number of languages in each subnode is in parantheses (). The Italian word tokenizer was propagated to other languages in the Italo-Dalmatian subfamily, while Spanish was propagated up the tree from the Western branch, given that it is a higher resource language than Italian. Latin has its own native tokenizer. Word tokenizers are propagated all the way to the first level, but not to the root or across top level subfamilies.

### A.2 Canary Languages

While our corpus and pipeline covers more than a thousand languages, we perform in-depth evaluations of the following subset of languages:

| Language | Family | Script | Resource Availability |
|---|---|---|---|
| Arabic | Afro-Asiatic | Arabic | Medium |
| Chinese | Sino-Tibetan | Han | High |
| French | Indo-European (Italic) | Latin | High |
| Hindi | Indo-European (Indo-Iranian) | Devanagari | Medium |
| Russian | Indo-European (Balto-Slavic) | Cyrillic | High |
| Swahili | Niger-Congo | Latin | Low |
| Telugu | Dravidian | Telugu | Low |
| Thai | Kra-Dai | Thai | Medium |
| Turkish | Turkic | Latin | Medium |

Table 1: The 9 canary languages and their families, main script, and resource availability.

### A.3 Multilingual tokenizers comparison

Following Rust et al. (2021), we considered two metrics:

- **Subword fertility (sf)**: the average number of tokens per "real" text word. Measures how aggressively a tokenizer splits words. The theoretical minimum of 1 would mean the tokenizer vocabulary contains every single word from the reference text;
- **Proportion of continued words (pcw)**: the ratio of "real" text words encoded with 2 tokens or more. Measures how often a tokenizer splits words. A value of 0 means that the tokenizer never splits and 1 that it always splits.

We split each language's Wikipedia into individual words (see Section 2) and computed the two metrics using tokenizers from a variety of popular multilingual models: Mistral-7B-V3 (Jiang et al., 2023), Phi3 (Abdin et al., 2024), Llama3 (AI@Meta, 2024), Qwen2.5 (Yang et al., 2024), mT5 (Xue et al., 2021), Bigscience-Bloom (Workshop et al., 2023), Command-R (Cohere, 2024), Gemma (Gemma Team et al., 2024), and XGLM (Lin et al., 2022). We did not include tokenizers in our comparison if they had a vocabulary size over 256,000, as these would make the embedding layer consume a considerable number of paramaters: at the relatively small model scale we targeted for our experiments (around 1.5 billion parameters), this would force us to significantly reduce the number of model layers due to computational constraints.

Following inspection of the computed metrics in Appendix A.3, where we additionally show the average and worst-case (max), and lower is better for both metrics, we excluded tokenizers that showed very low subword fertility or proportion of continued words on at least one of our canary languages. The Mistral-7B-V3, Phi3, Command-R and Llama3 tokenizers all do not provide good coverage of Telugu. Additionally, while XGLM and mT5 both showed strong performance, they do not preserve whitespaces, and some characters (particularly for Chinese) would be encoded as "unknown token" ([UNK]). Ultimately, our tokenizer of choice was Gemma, a modern BPE tokenizer that performed slightly better than Bigscience-Bloom on average for our experimental setup.

| Tokenizer | Mistralv3 | Phi3 | Llama3 | Qwen2.5* | mT5 | Bloom | Cmd-R | Gemma | XGLM |
|---|---|---|---|---|---|---|---|---|---|
| Vocab size | 32,768 | 100,352 | 128,000 | 151,643 | 250,100 | 250,680 | 255,000 | 256,000 | 256,008 |
| No [UNK] | ✓ | ✓ | ✓ | ✓ | ✗ | ✓ | ✓ | ✓ | ✗ |
| English sf | 1.45 | 1.40 | 1.40 | 1.47 | 1.52 | 1.42 | 1.35 | **1.31** | 1.34 |
| English pcw | 0.23 | 0.28 | 0.28 | 0.29 | 0.45 | 0.31 | 0.22 | **0.19** | 0.28 |
| Chinese sf | 3.03 | 2.30 | 1.60 | 1.44 | 2.29 | **1.29** | 1.35 | 1.43 | 2.21 |
| Chinese pcw | 0.95 | 0.58 | 0.43 | 0.31 | 0.91 | **0.23** | 0.25 | 0.32 | 0.82 |
| French sf | 1.69 | 1.74 | 1.73 | 1.76 | 1.71 | 1.49 | 1.50 | 1.50 | **1.45** |
| French pcw | 0.40 | 0.47 | 0.47 | 0.47 | 0.55 | 0.35 | 0.35 | **0.34** | 0.35 |
| Russian sf | 2.42 | 2.99 | 2.34 | 2.50 | 1.96 | 2.86 | 1.99 | 2.05 | **1.68** |
| Russian pcw | 0.59 | 0.66 | 0.62 | 0.64 | 0.73 | 0.63 | 0.56 | 0.57 | **0.50** |
| Turkish sf | 3.18 | 2.63 | 2.32 | 2.55 | 1.99 | 2.59 | 2.13 | 2.22 | **1.72** |
| Turkish pcw | 0.74 | 0.70 | 0.68 | 0.70 | 0.73 | 0.67 | 0.64 | 0.66 | **0.53** |
| Arabic sf | 4.76 | 3.72 | 2.32 | 2.23 | 2.10 | 1.86 | 2.16 | 2.19 | **1.72** |
| Arabic pcw | 0.92 | 0.86 | 0.74 | 0.67 | 0.79 | 0.60 | 0.68 | 0.69 | **0.52** |
| Thai sf | 4.87 | 3.80 | 2.18 | 2.44 | 1.99 | 3.96 | 4.01 | 1.92 | **1.78** |
| Thai pcw | 0.93 | 0.85 | 0.66 | 0.64 | 0.68 | 0.86 | 0.87 | **0.46** | 0.53 |
| Hindi sf | 4.99 | 4.60 | 2.71 | 3.98 | 2.02 | 1.59 | 3.39 | 2.22 | **1.52** |
| Hindi pcw | 0.91 | 0.90 | 0.81 | 0.86 | 0.69 | 0.39 | 0.80 | 0.60 | **0.33** |
| Swahili sf | 2.30 | 2.09 | 2.07 | 2.16 | 1.78 | 1.72 | 1.95 | 1.84 | **1.54** |
| Swahili pcw | 0.63 | 0.62 | 0.62 | 0.63 | 0.62 | 0.52 | 0.59 | 0.53 | **0.42** |
| Telugu sf | 9.83 | 10.11 | 10.11 | 8.41 | 2.44 | **2.10** | 9.74 | 3.51 | 2.24 |
| Telugu pcw | 0.79 | 0.76 | 0.76 | 0.77 | 0.86 | **0.59** | 0.78 | 0.74 | 0.69 |
| Max sf | 9.83 | 10.11 | 10.11 | 8.41 | 2.44 | 3.96 | 9.74 | 3.51 | **2.24** |
| Max pcw | 0.95 | 0.90 | 0.81 | 0.86 | 0.91 | 0.86 | 0.87 | **0.74** | 0.82 |
| Avg sf | 4.12 | 3.78 | 3.04 | 3.05 | 2.03 | 2.16 | 3.14 | 2.10 | **1.76** |
| Avg pcw | 0.76 | 0.71 | 0.64 | 0.63 | 0.73 | 0.54 | 0.61 | 0.55 | **0.52** |

Table 2: Multilingual Tokenizers Comparison on Wikipedia. * denotes tokenizers that were not originally available when we first ran this comparison. [UNK] is the unknown token: mT5 and XGLM are unable to encode some characters, particularly for Chinese. Avg is the average across all languages, and Max the maximum (worst-case) across all languages. Lower is better for all rows.

## A.4 Model architecture and training

| Parameter | Value |
|---|---|
| Architecture | Llama |
| Number of attention heads | 32 |
| Number of hidden layers | 14 |
| Number of key-value heads | 32 |
| RMS Norm epsilon | 1e-05 |
| d_model | 2048 |
| Tied word embeddings | True |
| Embedding size | 256008 |
| Total number of parameters | 1.46B |
| Random initialization std | 0.02 |
| Tokenizer | Gemma |

Table 3: **Architecture configuration** for all models

| Parameter | 29BT | 100BT | 350BT |
|---|---|---|---|
| Data parallelism (dp) | 64 | 56 | 64 |
| Tensor parallelism (tp) | 1 | 1 | 1 |
| Pipeline parallelism (pp) | 1 | 1 | 1 |
| Sequence length | 2048 | 2048 | 2048 |
| Batch size (samples) | 1024 | 840 | 1280 |
| Batch size (tokens) | 2097152 | 1720320 | 2621440 |

Table 4: **Training settings** for the 3 training scales we consider: 29, 100 and 350 billion tokens. For 100BT and 350BT, we compute critical batch size based on DeepSeek-AI et al. (2024)

| Parameter | 29BT | 100BT | 350BT |
|---|---|---|---|
| Adam beta1 | 0.9 | 0.9 | 0.9 |
| Adam beta2 | 0.95 | 0.95 | 0.95 |
| Adam epsilon | 1.0e-8 | 1.0e-8 | 1.0e-8 |
| Gradient clipping | 1.0 | 1.0 | 1.0 |
| Weight decay | 0.1 | 0.1 | 0.1 |
| Learning rate | 3e-4 | 8e-4 | 7e-4 |
| Total train steps | 14000 | 59000 | 134000 |
| Warmup steps | 500 | 2950 (5%) | 6700 (5%) |
| Warmup style | linear | linear | linear |
| Decay steps | 13500 | 11800 (20%) | 26800 (20%) |
| Decay starting step | 500 | 47200 | 107200 |
| Decay style | cosine | linear | linear |
| Minimum decay LR | 3.0e-5 | 0 | 0 |

Table 5: **Optimizer settings** for the 3 training scales we consider: 29, 100 and 350 billion tokens. For 100BT and 350BT, we train with a constant learning rate until the last 20% of steps (computed following DeepSeek-AI et al. (2024)), so that the resulting models can easily undergo continued pretraining.

### A.5 Evaluation details

#### A.5.1 Task selection criteria

As noted in Section 3.3, we define precise quantitative criteria for each of the properties of the early-signal task. To compute each criterion, we only use models trained on available reference datasets for given language (see Section 3.2), denoted as $M$. Every task in our final selection had to satisfy all of the following criteria requirements:

**Monotonicity.** To assess *Monotonicity* of a task, we compute the average Spearman rank correlation between the evaluation steps and the corresponding model scores. For a given model $m$, let the score at step $s$ be denoted $m(s)$. The average monotonicity across all models is then defined as:

$$\bar{\rho} = \frac{1}{|M|} \sum_{m \in M} \rho \left( [s_0, s_1, \ldots, s_n], [m(s_0), m(s_1), \ldots, m(s_n)] \right)$$

Here, the Spearman correlation $\rho(x, y)$ between sequences $x = [x_1, \ldots, x_n]$ and $y = [y_1, \ldots, y_n]$ is computed as:

$$\rho(x, y) = 1 - \frac{6 \sum_{i=1}^{n} d_i^2}{n(n^2 - 1)}$$

where $d_i = \text{rank}(x_i) - \text{rank}(y_i)$ is the rank difference for element $i$, and $n$ is the number of evaluation steps. We consider a task to meet the monotonicity criterion if:

$$\bar{\rho} \geq 0.5$$

**Signal-to-Noise Ratio (SNR).** Inspired by Madaan et al. (2024) we estimate how robust is a task to training noise, by computing its *Signal-to-Noise Ratio (SNR)* using four models trained on unfiltered CommonCrawl data under different random seeds:

- **seed-3:** Trained on a random subset with data and model seed set to 3
- **seed-4:** Trained on same subset as seed 3, with data and model seed set to 4
- **seed-5:** Trained on a different random subset with data and model seed set to 5
- **seed-6:** Trained on the same subset as 5 with data seed = 6 and model seed = 42

We refer to this set of four models as $MC$. For each evaluation step $s$, we define the mean score (signal) as:

$$\mu_s^* = \frac{1}{|MC|} \sum_{m \in MC} m(s)$$

and the standard deviation (noise) as:

$$\sigma_s = \sqrt{\frac{1}{|MC|} \sum_{m \in MC} (m(s) - \mu_s^*)^2}$$

The overall task SNR is then the average ratio of signal to noise across all $n$ training steps:

$$\text{SNR} = \frac{1}{n} \sum_{s=0}^{n} \frac{\mu_s^*}{\sigma_s}$$

We chose the minimum required SNR to 20, with the exception of generative tasks, which we found to be considerably "noisier" in general, but we wanted to have at least one generative task per language. Generative tasks are quite relevant in a multilingual context as they provide insights into how the model behaves when prompted to generate unconstrained, i.e., without a limited set of answer options. Models trained in multiple languages can sometimes exhibit high scores in multiple choice tasks but reply in the wrong language for generative tasks ("accidental translation"), or otherwise lack fluency (Xue et al., 2021).

**Non-Random Performance.** To assess that non-zero task results are not just a consequence of random noise, we look at the best score at the last evaluation step among models from $M$. We first compute the maximum improvement over a random baseline $b$:

$$\max_d = \max_{m \in M} (m(n) - b)$$

We then estimate the variance at the end of training using the standard deviation (from previous calculation) averaged over the last 5 steps:

$$\sigma_{\text{end}} = \frac{1}{5} \sum_{s=n-4}^{n} \sigma_s$$

Finally, The non-randomness score is defined as the ratio of max improvement to this terminal variance:

$$\text{non\_randomness} = \frac{\max_d}{\sigma_{\text{end}}}$$

A task satisfies the non-randomness criterion if:

$$\text{non\_randomness} \geq 3$$

**Ordering Consistency** To compute how consistently models are ordered as training progresses, we calculate the average Kendall Tau-a between model rankings at consecutive steps in the second half of training. We ignore the first 15 billion tokens, as we are interested in this property at a later stage of training, and in the first half, we found the ordering to be very inconsistent, skewing the overall score. First, we define Kendall Tau-a of model ranking as:

$$\tau_a(x, y) = \frac{C - D}{\binom{n}{2}}$$

where $C$ and $D$ are the number of concordant and discordant pairs between the rankings $x$ and $y$ of the model scores at steps $s_i$ and $s_{i+1}$. The overall consistency is:

$$\text{ordering\_consistency} = \frac{1}{|P|} \sum_{(s_i, s_{i+1}) \in P} \tau_a \left( r(s_i), r(s_{i+1}) \right)$$

where $P$ is the set of consecutive step pairs in the latter half of the training, and $r(s)$ is the ranking of model scores in step $s$.

While we first considered using the criteria for selection, we could not determine a reliable threshold for the criterion and therefore only use it for observational reasons.

### A.5.2 Metrics and Formulation

For non-generative tasks, we compute accuracies using **Cloze Formulation** (CF, completing with the full option text) in place of the more commonly used Multi-Choice Formulation

(MCF, completing with A/B/C/D), as previous work has shown that MCF has random performance in the early stages of training (Gu et al., 2025; Li et al., 2024b).

Additionally, since all models that we compare use the same tokenizer, we **normalize answer log-probabilities based on token count** instead of number of characters, and use pointwise mutual information (**PMI**) (Gu et al., 2025) for more difficult tasks such as AGIEval (Zhong et al., 2023b) or translated versions of MMLU (Hendrycks et al., 2021). For these tasks, we use the **F1-score** of overlapping words, as it is generally less noisy and more resilient to small changes in the generations than exact matching (which in turn might be more appropriate for math related tasks, which we do not evaluate on).

### A.5.3 List of selected evaluation tasks for canary languages

| Task | Type | Metric | Mono | SNR | Rand | Order |
|------|------|--------|------|-----|------|-------|
| Belebele (Almazrouei et al., 2023) | RC | Acc (Char) | 0.61 | 58.23 | 14.67 | 0.13 |
| ArabicMMLU (Koto et al., 2024) | GK | Acc (PMI) | 0.81 | 80.00 | 18.28 | 0.91 |
| X-CSQA (Lin et al., 2021a) | RES | Acc (PMI) | 0.65 | 33.44 | 11.13 | 0.91 |
| Alghafa: MCQ Exams (Almazrouei et al., 2023) | GK | Acc (Token) | 0.51 | 35.49 | 8.89 | 0.61 |
| Alghafa: SOQAL (Almazrouei et al., 2023) | RC | Acc (Token) | 0.74 | 46.22 | 33.78 | 0.11 |
| Alghafa: ARC Easy (Leaderboard, 2024) | GK | Acc (Token) | 0.74 | 76.58 | 35.41 | 0.91 |
| Okapi: Hellaswag (Lai et al., 2023) | NLU | Acc (Token) | 0.80 | 43.05 | 12.01 | 0.97 |
| OALL2024: PIQA (Leaderboard, 2024) | RES | Acc (Token) | 0.81 | 69.34 | 7.69 | 0.71 |
| OALL2024: RACE (Leaderboard, 2024) | RC | Acc (Token) | 0.82 | 66.01 | 18.22 | 0.43 |
| OALL2024: SCIQ (Leaderboard, 2024) | GK | Acc (Token) | 0.80 | 74.06 | 32.87 | 0.70 |
| X-CODAH (Lin et al., 2021a) | RES | Acc (Token) | 0.75 | 24.80 | 8.50 | 0.31 |
| X-Story Cloze (Lin et al., 2021b) | NLU | Acc (Token) | 0.87 | 93.20 | 9.76 | 0.83 |
| ARCD (Mozannar et al., 2019) | GK | F1 | 0.83 | 28.28 | 35.58 | 0.83 |
| MLQA (Lewis et al., 2020) | RC | F1 | 0.86 | 17.27 | 24.83 | 0.87 |
| Tydiqa (Clark et al., 2020) | RC | F1 | 0.86 | 27.17 | 55.07 | 0.94 |

Table 6: Selected tasks for Arabic satisfying the early-signal conditions: Monotonicity (Mono), Signal-to-noise ratio (SNR), Non-Randomness (Rand) and Ordering Consistency (Order)

| Task | Type | Metric | Mono | SNR | Rand | Order |
|------|------|--------|------|-----|------|-------|
| AGIEval (ZH subset) (Zhong et al., 2023a) | GK | Acc (PMI) | 0.46 | 98.82 | 15.86 | 0.86 |
| X-CSQA (Lin et al., 2021a) | RES | Acc (PMI) | 0.83 | 25.63 | 10.09 | 0.89 |
| Belebele (Bandarkar et al., 2024) | RC | Acc (Token) | 0.51 | 74.30 | 15.36 | 0.70 |
| C3 (Sun et al., 2020) | RC | Acc (Token) | 0.87 | 72.89 | 36.01 | 0.66 |
| C-Eval (Huang et al., 2023) | GK | Acc (Token) | 0.75 | 50.20 | 8.04 | 0.53 |
| CMMLU (Li et al., 2024a) | GK | Acc (Token) | 0.91 | 117.92 | 21.93 | 0.96 |
| Okapi: Hellaswag (Lai et al., 2023) | NLU | Acc (Token) | 0.87 | 70.60 | 21.95 | 0.97 |
| M3Exam (Zhang et al., 2023) | GK | Acc (Token) | 0.74 | 36.02 | 8.75 | 0.67 |
| X-CODAH (Lin et al., 2021a) | RES | Acc (Token) | 0.66 | 32.65 | 14.72 | 0.66 |
| X-COPA (Ponti et al., 2020) | RES | Acc (Token) | 0.80 | 77.20 | 15.06 | 0.69 |
| X-Story Cloze (Lin et al., 2021b) | NLU | Acc (Token) | 0.87 | 79.20 | 15.57 | 0.84 |
| X-Winograd (Muennighoff et al., 2022) | NLU | Acc (Token) | 0.88 | 102.87 | 21.83 | 0.86 |
| Chinese SQuAD (Pluto-Junzeng, 2019) | RC | F1 | 0.85 | 27.71 | 27.40 | 0.90 |
| CMRC (Cui et al., 2018) | RC | F1 | 0.91 | 25.33 | 34.43 | 0.67 |
| MLQA (Lewis et al., 2020) | RC | F1 | 0.91 | 23.76 | 20.40 | 0.86 |

Table 7: Selected tasks for Chinese satisfying the early-signal conditions: Monotonicity (Mono), Signal-to-noise ratio (SNR), Non-Randomness (Rand) and Ordering Consistency (Order)

| Task | Type | Metric | Mono | SNR | Rand | Order |
|------|------|--------|------|-----|------|-------|
| Okapi: ARC (Lai et al., 2023) | GK | Acc (PMI) | 0.69 | 30.10 | 3.33 | 0.47 |
| Meta MMLU (Grattafiori et al., 2024) | GK | Acc (PMI) | 0.87 | 107.58 | 10.95 | 0.56 |
| X-CSQA (Lin et al., 2021a) | RES | Acc (PMI) | 0.83 | 30.50 | 11.01 | 0.76 |
| Belebele (Bandarkar et al., 2024) | RC | Acc (Token) | 0.85 | 33.68 | 5.65 | 0.39 |
| Okapi: Hellaswag (Lai et al., 2023) | NLU | Acc (Token) | 0.96 | 71.11 | 30.84 | 0.70 |
| X-CODAH (Lin et al., 2021a) | RES | Acc (Token) | 0.74 | 33.68 | 9.19 | 0.74 |
| FQuad (d'Hoffschmidt et al., 2020) | RC | F1 | 0.91 | 14.64 | 19.08 | 0.69 |
| Mintaka (Sen et al., 2022) | GK | F1 | 0.82 | 6.91 | 12.92 | 0.79 |

Table 8: Selected tasks for French satisfying the early-signal conditions: Monotonicity (Mono), Signal-to-noise ratio (SNR), Non-Randomness (Rand) and Ordering Consistency (Order)

| Task | Type | Metric | Mono | SNR | Rand | Order |
|------|------|--------|------|-----|------|-------|
| Meta MMLU (Grattafiori et al., 2024) | GK | Acc (PMI) | 0.68 | 97.78 | 9.13 | 0.33 |
| X-CSQA (Lin et al., 2021a) | RES | Acc (PMI) | 0.60 | 22.84 | 4.45 | 1.00 |
| Belebele (Bandarkar et al., 2024) | RC | Acc (Token) | 0.61 | 66.05 | 6.65 | 0.76 |
| Okapi: Hellaswag (Lai et al., 2023) | NLU | Acc (Token) | 0.87 | 47.47 | 16.35 | 1.00 |
| Okapi: ARC (Lai et al., 2023) | GK | Acc (Token) | 0.95 | 62.19 | 23.11 | 0.67 |
| X-CODAH (Lin et al., 2021a) | RES | Acc (Token) | 0.53 | 39.83 | 14.13 | 0.67 |
| X-Story Cloze (Lin et al., 2021b) | NLU | Acc (Token) | 0.74 | 87.75 | 8.39 | 1.00 |
| IndicQA (Singh et al., 2025) | RC | F1 | 0.94 | 13.20 | 12.20 | 0.81 |

Table 9: Selected tasks for Hindi satisfying the early-signal conditions: Monotonicity (Mono), Signal-to-noise ratio (SNR), Non-Randomness (Rand) and Ordering Consistency (Order)

| Task | Type | Metric | Mono | SNR | Rand | Order |
|------|------|--------|------|-----|------|-------|
| Okapi: ARC (Lai et al., 2023) | GK | Acc (PMI) | 0.55 | 35.17 | 3.76 | 0.53 |
| RUMMLU (Fenogenova et al., 2024) | GK | Acc (PMI) | 0.77 | 64.24 | 6.10 | 0.56 |
| X-CSQA (Lin et al., 2021a) | RES | Acc (PMI) | 0.73 | 38.45 | 16.03 | 0.71 |
| Belebele (Bandarkar et al., 2024) | RC | Acc (Token) | 0.81 | 61.97 | 19.26 | 0.71 |
| Okapi: Hellaswag (Lai et al., 2023) | NLU | Acc (Token) | 0.97 | 86.76 | 28.22 | 0.83 |
| Parus (Fenogenova et al., 2024) | RES | Acc (Token) | 0.93 | 81.06 | 24.61 | 0.67 |
| OpenBookQA (Fenogenova et al., 2024) | RES | Acc (Token) | 0.73 | 43.43 | 18.08 | 0.73 |
| X-CODAH (Lin et al., 2021a) | RES | Acc (Token) | 0.85 | 26.97 | 6.79 | 0.50 |
| X-Story Cloze (Lin et al., 2021b) | NLU | Acc (Token) | 0.93 | 66.81 | 12.04 | 0.84 |
| Sber SQuAD (Efimov et al., 2020) | RC | F1 | 0.89 | 9.93 | 10.85 | 0.84 |
| Tydiqa (Clark et al., 2020) | RC | F1 | 0.92 | 10.44 | 11.28 | 0.83 |
| X-QuAD (Artetxe et al., 2020a) | RC | F1 | 0.90 | 8.79 | 7.56 | 0.60 |

Table 10: Selected tasks for Russian satisfying the early-signal conditions: Monotonicity (Mono), Signal-to-noise ratio (SNR), Non-Randomness (Rand) and Ordering Consistency (Order)

| Task | Type | Metric | Mono | SNR | Rand | Order |
|------|------|--------|------|-----|------|-------|
| Okapi: ARC (Lai et al., 2023) | GK | Acc (Token) | 0.88 | 60.69 | 6.32 | - |
| Belebele (Bandarkar et al., 2024) | RC | Acc (Token) | 0.44 | 65.26 | 5.44 | - |
| M3Exam (Zhang et al., 2023) | GK | Acc (Token) | 0.63 | 34.82 | 3.52 | - |
| X-COPA (Ponti et al., 2020) | RES | Acc (Token) | 0.82 | 74.71 | 4.66 | - |
| X-Story Cloze (Lin et al., 2021b) | NLU | Acc (Token) | 0.86 | 130.08 | 20.54 | - |
| KenSWQuAD (Wanjawa et al., 2023) | RC | F1 | 0.91 | 12.95 | 12.43 | - |
| Tydiqa (Clark et al., 2020) | RC | F1 | 0.65 | 12.67 | 15.01 | - |

Table 11: Selected tasks for Swahili satisfying the early-signal conditions: Monotonicity (Mono), Signal-to-noise ratio (SNR), Non-Randomness (Rand) and Ordering Consistency (Order)

| Task | Type | Metric | Mono | SNR | Rand | Order |
|------|------|--------|------|-----|------|-------|
| Okapi: Hellaswag (Lai et al., 2023) | NLU | Acc (Token) | 0.82 | 56.06 | 7.84 | - |
| Okapi: MMLU (Lai et al., 2023) | GK | Acc (Token) | 0.92 | 148.57 | 4.11 | - |
| X-COPA (Ponti et al., 2020) | RES | Acc (Token) | 0.77 | 69.31 | 6.01 | - |
| X-Story Cloze (Lin et al., 2021b) | NLU | Acc (Token) | 0.67 | 108.25 | 8.02 | - |
| IndicQA (Singh et al., 2025) | RC | F1 | 0.72 | 12.39 | 9.65 | - |

Table 12: Selected tasks for Telugu satisfying the early-signal conditions: Monotonicity (Mono), Signal-to-noise ratio (SNR), Non-Randomness (Rand) and Ordering Consistency (Order)

| Task | Type | Metric | Mono | SNR | Rand | Order |
|------|------|--------|------|-----|------|-------|
| Meta MMLU (Grattafiori et al., 2024) | GK | Acc (PMI) | 0.54 | 93.51 | 6.42 | 0.60 |
| Belebele (Bandarkar et al., 2024) | RC | Acc (Token) | 0.63 | 53.88 | 13.65 | 0.66 |
| Translated Hellaswag (Patteera, 2023) | NLU | Acc (Token) | 0.69 | 52.78 | 11.51 | 0.53 |
| M3Exam (Zhang et al., 2023) | GK | Acc (Token) | 0.75 | 45.32 | 4.24 | 0.50 |
| ThaiQA (Trakultaweekoon et al., 2019) | RC | F1 | 0.90 | 20.39 | 15.92 | 0.66 |
| X-QuAD (Artetxe et al., 2020a) | RC | F1 | 0.90 | 17.45 | 20.07 | 0.80 |

Table 13: Selected tasks for Thai satisfying the early-signal conditions: Monotonicity (Mono), Signal-to-noise ratio (SNR), Non-Randomness (Rand) and Ordering Consistency (Order)

| Task | Type | Metric | Mono | SNR | Rand | Order |
|------|------|--------|------|-----|------|-------|
| TR Leaderboard: ARC (Alhajar, 2024) | GK | Acc (Char) | 0.91 | 49.33 | 21.32 | 0.79 |
| Belebele (Bandarkar et al., 2024) | RC | Acc (Char) | 0.50 | 47.97 | 5.93 | 0.09 |
| Exams (Hardalov et al., 2020) | GK | Acc (Char) | 0.78 | 31.73 | 5.96 | 0.33 |
| Okapi: Hellaswag (Lai et al., 2023) | NLU | Acc (Char) | 0.95 | 58.56 | 21.45 | 0.90 |
| X-COPA (Ponti et al., 2020) | RES | Acc (Char) | 0.61 | 81.18 | 11.43 | 0.66 |
| TR Leaderboard: MMLU (Alhajar, 2024) | GK | Acc (PMI) | 0.81 | 95.48 | 12.60 | 0.61 |
| THQuAD (Soygazi et al., 2021) | RC | F1 | 0.93 | 17.06 | 20.03 | 0.60 |
| X-QuAD (Artetxe et al., 2020a) | RC | F1 | 0.92 | 26.33 | 28.74 | 0.73 |

Table 14: Selected tasks for Turkish satisfying the early-signal conditions: Monotonicity (Mono), Signal-to-noise ratio (SNR), Non-Randomness (Rand) and Ordering Consistency (Order)

### A.6 Language Identification

#### A.6.1 Classifier choice

While Transformer-based LID classifiers exist (Bapna et al., 2022), they are too slow and expensive to run at a large scale. Most commonly used LID classifiers are simple models based on character level n-grams like CLD3 (Salcianu et al., 2018) (107 supported languages), used in mC4, or classifiers following the fastText architecture (Joulin et al., 2016), such as FT176 (176 languages) used in CC-100 (Wenzek et al., 2020; Conneau et al., 2020), and CulturaX (Nguyen et al., 2024), as well as in many English-only datasets (Soldaini et al., 2024; Penedo et al., 2023); OpenLID (Burchell et al., 2023) (193 languages), used in HPLT2 (Burchell et al., 2025); and the recent GlotLID (Kargaran et al., 2023) (1880 languages).

We used the GlotLID (Kargaran et al., 2023), specifically version V3—the latest available at the time of our experiments (Kargaran et al., 2024). This LID classifier covers a large number of languages and addresses some common issues in LID classifiers:

- Its large language coverage can reduce "out-of-model cousin" errors (Caswell et al., 2020; Bapna et al., 2022), where unsupported languages can be misclassified as a closely related supported language

- It explicitly distinguishes scripts (Latin, Arabic, Cyrillic, etc), improving detection for languages that support multiple scripts

- It provides different labels based on script e.g. 'arb_Arab' is Standard Arabic in Arabic script, while 'arb_Latn' is Standard Arabic in Latin script, allowing us to tailor the filtering to each script

- It includes an "UND" label for non supported scripts, so that languages that use them aren't misclassified as supported languages

- Includes specific labels trained on "noise" documents, such as text decoded with the wrong encoding, binary content, or misrendered PDFs, preventing it from being classified as a natural text language

In Fig. 5 we present a comparison between GlotLID and FT176, without any threshold filtering.

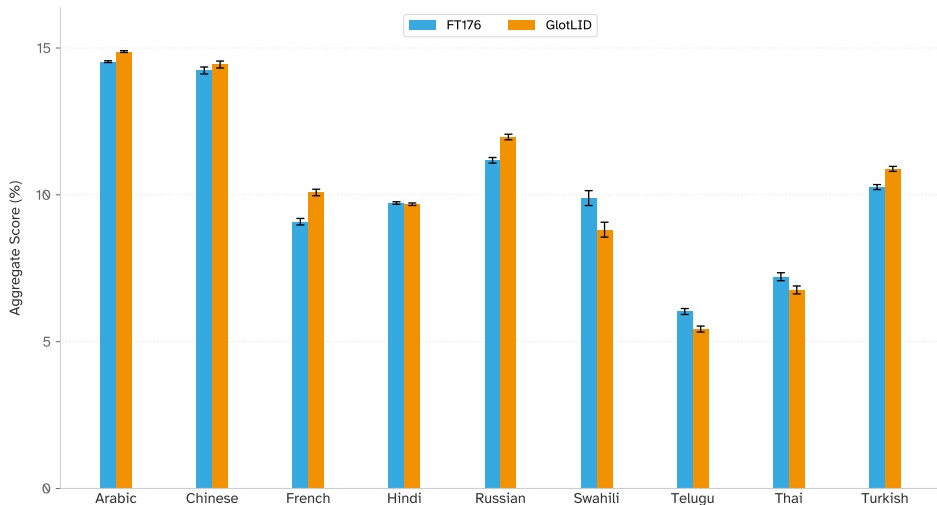

Figure 5: **FT176 vs GlotLID** without any threshold filtering applied to either classifier. While GlotLID seems to outperform in higher resource languages, FT176 performs slightly better on lower resource languages. However, GlotLID supports a considerably larger number of (lower-resource) languages.

*A.6.2   Confidence Threshold*

For each language, we set thresholds at regular removal intervals (threshold to remove 5% of data, 10% of data, etc) and at other values of interest (e.g., 0.7, 0.9). We then train models on 30 billion tokens of the filtered data using each threshold and evaluate the resulting models. In Tables 16 to 24, the highest scoring range of thresholds are marked in bold. Our formula to automatically set thresholds based on the mean and standard deviation of each language's confidence scores distribution selects values within the highest scoring range for all languages except Chinese and Hindi (Table 15).

| Language | Minimum | Maximum | Formula | Value in range |
|----------|---------|---------|---------|----------------|
| Arabic | 0.883 | 0.9 | 0.8812 | ✓ |
| Chinese | 0.895 | 0.937 | 0.7415 | ✗ |
| French | 0.750 | 0.932 | 0.8195 | ✓ |
| Hindi | 0.483 | 0.557 | 0.6827 | ✗ |
| Russian | 0.8 | - | 0.9 | ✓ |
| Swahili | 0.186 | 0.544 | 0.3 | ✓ |
| Telugu | 0.701 | 0.701 | 0.7002 | ✓ |
| Thai | 0.9 | 0.961 | 0.9 | ✓ |
| Turkish | 0.866 | - | 0.8753 | ✓ |

Table 15: Minimum and Maximum refer to the highest performing threshold range endpoints. Formula is the value defined by our threshold setting formula, which we ultimately use for all languages.

| | Model 1 | Model 2 | Model 3 | Model 4 | Model 5 |
|--|---------|---------|---------|---------|---------|
| Threshold | 0.000 | 0.700 | 0.883 | 0.900 | 0.968 |
| % Removed | 0.0% | 3.0% | 5.0% | 5.4% | 10.0% |
| Aggregate Score | 14.9% | 15.2% | **15.4%** | **16.1%** | 15.2% |

Table 16: Arabic Threshold Analysis

| | Model 1 | Model 2 | Model 3 | Model 4 | Model 5 | Model 6 | Model 7 |
|--|---------|---------|---------|---------|---------|---------|---------|
| Threshold | 0.000 | 0.214 | 0.429 | 0.678 | 0.806 | 0.895 | 0.937 |
| % Removed | 0.0% | 5.0% | 10.0% | 15.0% | 20.0% | 25.0% | 30.0% |
| Aggregate Score | 14.4% | 14.4% | 14.8% | 14.8% | 14.4% | **15.1%** | **15.1%** |

Table 17: Chinese Threshold Analysis

|  | Model 1 | Model 2 | Model 3 | Model 4 | Model 5 | Model 6 | Model 7 |
|---|---|---|---|---|---|---|---|
| Threshold | 0.000 | 0.467 | 0.723 | 0.750 | 0.800 | 0.867 | 0.932 |
| % Removed | 0.0% | 5.0% | 10.0% | 10.7% | 12.2% | 15.0% | 20.0% |
| Aggregate Score | 10.1% | 10.5% | 10.5% | **11.1%** | **10.7%** | **11.0%** | **11.2%** |

Table 18: French Threshold Analysis

|  | Model 1 | Model 2 | Model 3 | Model 4 | Model 5 |
|---|---|---|---|---|---|
| Threshold | 0.000 | 0.483 | 0.557 | 0.616 | 0.669 |
| % Removed | 0.0% | 5.0% | 10.0% | 15.0% | 20.0% |
| Aggregate Score | 9.7% | **9.8%** | **9.8%** | 9.2% | 9.3% |

|  | Model 6 | Model 7 | Model 8 | Model 9 | Model 10 | Model 11 |
|---|---|---|---|---|---|---|
| Threshold | 0.714 | 0.752 | 0.786 | 0.815 | 0.840 | 0.862 |
| % Removed | 25.0% | 30.0% | 35.0% | 40.0% | 45.0% | 50.0% |
| Aggregate Score | 9.5% | 9.7% | 9.0% | 9.3% | 9.2% | 8.7% |

Table 19: Hindi Threshold Analysis

|  | Model 1 | Model 2 | Model 3 | Model 4 |
|---|---|---|---|---|
| Threshold | 0.000 | 0.750 | 0.800 | 0.918 |
| % Removed | 0.0% | 2.5% | 2.9% | 5.0% |
| Aggregate Score | 12.0% | 12.1% | **12.4%** | **12.7%** |

Table 20: Russian Threshold Analysis

|  | Model 1 | Model 2 | Model 3 | Model 4 | Model 5 | Model 6 | Model 7 | Model 8 |
|---|---|---|---|---|---|---|---|---|
| Threshold | 0.000 | 0.075 | 0.098 | 0.132 | 0.167 | 0.186 | 0.300 | 0.544 |
| % Removed | 0.0% | 5.0% | 10.0% | 20.0% | 30.0% | 50.0% | 64.2% | 70.0% |
| Aggregate Score | 8.8% | 9.7% | 8.7% | 8.8% | 9.2% | **10.9%** | **10.9%** | **11.6%** |

Table 21: Swahili Threshold Analysis

|  | Model 1 | Model 2 | Model 3 | Model 4 | Model 5 | Model 6 | Model 7 | Model 8 |
|---|---|---|---|---|---|---|---|---|
| Threshold | 0.000 | 0.207 | 0.262 | 0.297 | 0.515 | 0.600 | 0.701 | 0.996 |
| % Removed | 0.0% | 5.0% | 10.0% | 15.0% | 20.0% | 22.4% | 25.0% | 30.0% |
| Aggregate Score | 5.4% | 5.0% | 5.4% | 5.1% | 5.2% | 5.3% | **5.9%** | 5.4% |

Table 22: Telugu Threshold Analysis

|  | Model 1 | Model 2 | Model 3 | Model 4 |
|---|---|---|---|---|
| Threshold | 0.000 | 0.800 | 0.900 | 0.961 |
| % Removed | 0.0% | 2.7% | 3.5% | 5.0% |
| Aggregate Score | 6.8% | 6.2% | **6.8%** | **6.9%** |

Table 23: Thai Threshold Analysis

|  | Model 1 | Model 2 | Model 3 | Model 4 | Model 5 | Model 6 | Model 7 |
|---|---|---|---|---|---|---|---|
| Threshold | 0.000 | 0.704 | 0.724 | 0.750 | 0.800 | 0.866 | 0.932 |
| % Removed | 0.0% | 5.0% | 6.1% | 6.6% | 7.7% | 10.0% | 13.9% |
| Aggregate Score | 10.9% | 10.3% | 9.3% | 10.2% | 10.5% | **11.5%** | **11.4%** |

Table 24: Turkish Threshold Analysis

### A.7  Filtering

#### A.7.1  Stopwords

As mentioned in Section 4.4.1, we analyzed word frequencies in our reference datasets (Wikipedia) using our word tokenizers to identify the most frequently occurring words.

After counting word occurrences directly on the raw data of our reference datasets, we noticed that some stopwords were actually non-alphabetic symbols or numbers rather than meaningful words. To refine the list, we removed all numbers and symbols. If fewer than eight stopwords (eight being the number of stopwords in the original Gopher English stopword list (Rae et al., 2022)) remained after this filtering, we lowered the frequency threshold to increase the number of stopwords and ensure sufficient stopword coverage.

When analyzing English stopwords from the Wikipedia reference dataset, we found that the original Gopher quality filter did not necessarily select the most frequent words. This suggested a different selection criterion had been used. However, since our method is scalable across languages and performs well in experiments, we adopted it as our approach and collected stopwords for each language supported by GlotLID (on Wikipedia when available, and on GlotLID-Corpus for languages that do not have their own Wikipedia).

When reviewing the languages with the largest amount of data after LID and stopwords filtering, we noticed that some low resource languages had an unexpectedly large amount of data. For example, Dagbani, a language from the Niger–Congo family with around 1 million native speakers and low internet presence, ended up with a large amount (2TB) of text data after language filtering. Through manual inspection, we found that most of this data was misclassified English and German. We had expected that the stopwords filter would remove most of this non-Dagbani content; however, the filter removed very little. Inspecting the list of Dagbani stopwords revealed a high amount of English words (shown in bold):

**the**, ni, **of**, **a**, **in**, ka, **and**, o, **be**, daa, **to**, di, n, nyɛla, **or**, **is**

Through further investigation, we found that many other languages had English stopwords in their list. We traced this issue to the Wikipedias for lower resource languages, where many articles are directly copied untranslated from the English wikipedia (for later translation) and some boilerplate/meta pages exist in the original English. As language classifiers are often trained on Wikipedia, this may explain why English data is mislabeled as these lowe-resource languages in the first place.

We "cleaned" Wikipedias by: a) removing the notes and references sections, which sometimes are in other languages and follow a very specific format; b) dropping articles where the most common script doesn't match what we expected for the language; c) dropping articles where our language classifier predicted English with above 70% confidence.
We then recomputed stopwords on this new clean version of Wikipedia, which resulted in a >99% removal rate when filtering Dagbani data using the updated stopwords:

ni, ka, o, daa, di, n, nyɛla, din, ti, bɛ, be, nyɛ, maa

### A.7.2 Filtering thresholds

**Filtering details**   We employ the following filters from FineWeb with fixed thresholds for all languages, only changing the way "words" are defined depending on each language's word-level tokenizer (Section 2):

- **FineWeb Quality filters**: ratio of characters in duplicate lines $\leq 0.1$;
- **Gopher Quality filters**: $50 \leq$ #words $\leq 100000$; ratio of symbols to #words $\leq 0.1$; ratio of bullet points to #lines $\leq 0.9$; ratio of ellipsis to #lines $\leq 0.3$, stop words in document $\geq 2$ (with stopwords determined following Section 4.4.1);

We *tune* the following filters with the different adaptation methods we consider:

- **FineWeb Quality filters**: maximum ratio of lines not ending with punctuation; maximum ratio of #lines to #words
- **Gopher Quality filters**: maximum average word length; minimum average word length; maximum ratio of non alphabetic words;
- **Gopher Repetition filters**: fraction of duplicate lines, fraction of characters taken up by the most common 2-, 3- and 4-grams; fraction of characters taken up by every single repeated 5-, 6-, 7-, 8-, 9-, and 10-gram

Results from training models on data obtained by applying the different adaptations methods to each group filters can be seen in Table 25. We **select the best performing method for each filter group** (marked in bold) for our pipeline. We also show the average removal rates across languages of each method in Table 26.

| Filter | | | | cc | | | | wiki | | |
|--------|----------|---------|--------|---------|----------|-------|--------|---------|----------|-------|
| Group | Baseline | English | 10Tail | MeanStd | MedRatio | Quant | 10Tail | MeanStd | MedRatio | Quant |
| **fwq** | 7.00 | - | - | 5.22 | 4.00 | 4.33 | **3.00** | 5.00 | 3.89 | 3.56 |
| **goq** | 6.33 | - | 5.22 | - | 3.89 | 4.56 | 4.44 | 4.11 | 4.22 | **3.22** |
| **gor** | 6.22 | 4.22 | 3.33 | **2.22** | - | 4.11 | - | 3.89 | - | 4.00 |

Table 25: **Average ranks** by block and method across all languages. *Baseline* has no filtering, English is the default FineWeb English thresholds. We then compute each of the other 4 methods – 10Tail, MeanStd, MedRatio (MedianRatio), and Quantile (Quant) – on both Common Crawl (cc) data and on Wikipedia (wiki). Cells marked with - correspond to method-filter-group combinations that would remove over 75% of data with a single filter on at least one of the languages, or that would not remove anything at all. Lower ranks are better.

| Filter | | | cc | | | | wiki | | |
|--------|---------|--------|---------|----------|--------|--------|---------|----------|--------|
| Group | English | 10Tail | MeanStd | MedRatio | Quant | 10Tail | MeanStd | MedRatio | Quant |
| **fwq** | - | - | 36.81% | 38.03% | 33.82% | 40.35% | 38.04% | 37.61% | 44.31% |
| **goq** | - | 41.42% | - | 47.09% | 45.14% | 49.23% | 47.58% | 46.90% | 46.81% |
| **gor** | 26.39% | 29.53% | 25.63% | - | 26.08% | - | 24.79% | - | 26.50% |

Table 26: **Average removal rates** by method across datasets. Values represent percentage of data filtered.

### A.7.3 Precision filtering lower resource languages

Language Identification precision computed on a balanced test set does not correspond to the precision on web crawled data, due to class imbalance between high- and low-resource languages. Precision on the crawled corpora can be calculated as in Caswell et al. (2020), where $x$ is the real proportion of the target language in the full web crawl:

$$\text{precision}_{\text{crawl}} = \frac{x \cdot \text{recall}}{x \cdot \text{recall} + (1 - x) \cdot \text{fpr}}$$

For low-resource languages (low $x$), a low false positive rate (fpr) is crucial, as higher false positives significantly reduce precision. For high-resource languages, web presence is high, so false positives are less critical.

If a low-resource language is sufficiently distinct from high-resource languages, the false positive rate will often be low. However, if a closely related high-resource language exists, the high-resource language may be misclassified as low-resource. In such scenarios, n-gram-based LID fails because common n-grams lead to misclassification of high-resource language sentences as low-resource.

**Wordlist filtering**    To maintain high precision for low-resource languages after LID, wordlist filtering is suggested to retain in-language documents (Caswell et al., 2020; Bapna et al., 2022).

To build such wordlists, we propose a simple approach: only consider tokens whose affinity exceeds a high threshold (we use $\gamma = 0.85$) for each language. The affinity of a token $t$ in language $l$ is defined as:

$$\text{Affinity}(t, l) = \frac{f_{t,l}}{\sum_{l' \in \mathcal{L}} f_{t,l'}}$$

where $f_{t,l}$ is the raw count of token $t$ in language $l$, and $\sum_{l' \in \mathcal{L}} f_{t,l'}$ is the total count of token $t$ across all languages in the set $\mathcal{L}$. A text labeled as a low-resource language $l$ is considered in-language if some of its words appear in the wordlist created for $l$; otherwise, it is considered contaminated. We used data from the GlotLID-Corpus (Kargaran et al., 2023) to create wordlists, applying the tokenizer specific to each language from Appendix A.1. For each language $l$, the same tokenizer (the tokenizer of language $l$) is used to compute $f_{t,l'}$ to ensure consistent separation of words.

We use wordlist filtering as an indicator of contamination, where the contamination score is defined as the percentage of documents removed by the filter. This helps identify languages with low quality for manual auditing. We select 10,000 random documents from each language and calculate the percentage of documents filtered. glk_Arab is one of the languages with the highest contamination score. We present the distribution of contamination scores for 1,900 languages for which we have wordlists in Fig. 6. The majority of the languages have their data in-language (non-contaminated). However, around a third of them have contamination scores above 10%.

**URL Whitelist**    Manual inspection of the filtering process revealed that some of the wordlists were too strict. This was the case of some English-based Pidgin languages, such as Nigerian Pidgin, for example, where the resulting wordlist was relatively short. To avoid excessive filtering caused by strict wordlists, we additionally kept documents removed by the wordlist filtering whose URLs contained specific terms related to the language (the language code, the name of the language, possibly top level domains for that region and/or names of regions where the language is spoken). For Nigerian Pidgin, this list contained the following words: "pcm", "pidgin", "naija", ".ng", "nigerianpidgin", "nigeria", and "nigerian". We show example URLs containing in-language content that had been removed by wordlist filtering that are then caught by the URL Whitelist in Table 27.

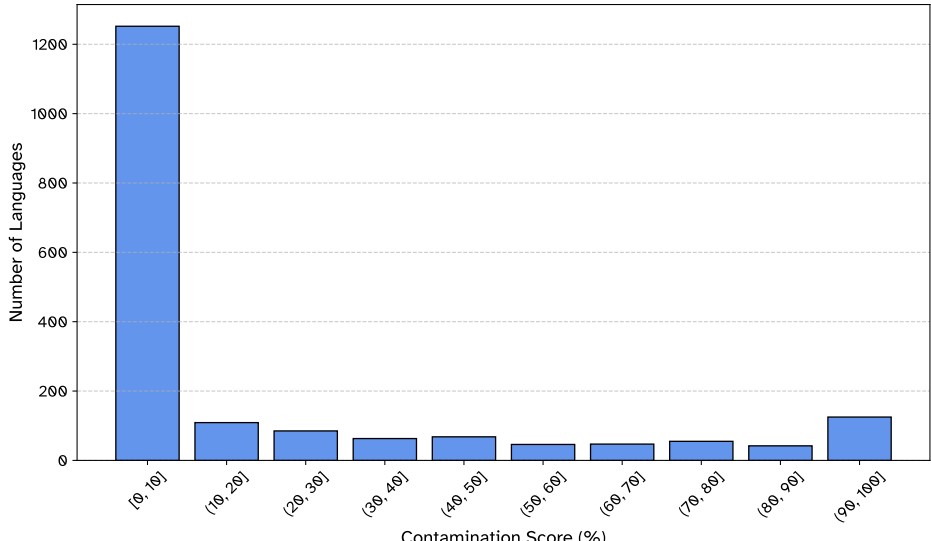

Figure 6: **Contamination scores for 1,900 languages via wordlist filtering.** The plot indicates that the majority of the languages have their data in-language (non-contaminated).

| URL | Matched Words |
|---|---|
| `http://www.supersport.com/football/nigeria-naija/news/121221/Uefa_don_ban_Malaga` | nigeria, nigerian, naija |
| `https://manutdinpidgin.com/2018/06/28/manchester-united-target-sergej-milinkovic-savic-don-react-ontop-the-transfer-rumour/` | pidgin |
| `https://pcm.wikipedia.org/wiki/Japan` | pcm |
| `https://www.bbc.com/pidgin/sport-43612518` | pidgin |

Table 27: Matched words for selected URLs in Nigerian Pidgin

**Filtering results** We audit three low-resource languages: glk_Arab, bar_Latn, and ary_Arab, by asking native speakers to manually label 2,000 randomly sampled documents as being in-language or not. The results of applying wordlist filtering with the URL Whitelist to these languages are shown in Table 28. Applying wordlist filtering maintains recall while improving precision for both glk_Arab and bar_Latn. However, for ary_Arab, the improvement is not very significant. This is because the training data for LID does not adequately represent ary_Arab. Precision could be increased further by requiring a certain fraction of the document to be contained in the wordlist (instead of just a single word), but this would require manual tuning and could result in a drop in recall.

| Language | Pre-filtering Precision | Filtering Recall | Filtering Precision |
|---|---|---|---|
| glk_Arab | 2.10% | 95.24% | 27.21% |
| bar_Latn | 69.45% | 97.77% | 94.90% |
| ary_Arab | 1.75% | 88.57% | 4.14% |

Table 28: Evaluation Results for wordlist filtering Based on the Audit

We publicly release our wordlists and code.[6]

---

[6]`https://github.com/huggingface/fineweb-2/tree/main/misc/precision_filtering`

## A.8 Improvement from each pipeline step

- **LID** Language Identification and threshold
- **LID + D** LID & global MinHash deduplication
- **LID + D + F** LID + D & heuristic filtering
- **FW2 (R)** LID + D + F & rehydration (deduplication informed upsampling)

| Task | Random Baseline | LID | | LID + D | | LID + D + F | | FW2 (R) | |
|---|---|---|---|---|---|---|---|---|---|
| | | Raw | Rescaled | Raw | Rescaled | Raw | Rescaled | Raw | Rescaled |
| Alghafa: MCQ Exams (GK) | 25.0 | **37.1** | **16.1** | 35.5 | 14.0 | 35.6 | 14.1 | 36.3 | 15.1 |
| Belebele (RC) | 25.0 | 32.1 | 9.4 | 31.4 | 8.5 | 32.6 | 10.1 | **33.9** | **11.9** |
| Alghafa: SOQAL (RC) | 20.0 | 64.0 | 55.0 | 62.3 | 52.9 | **67.9** | **59.9** | 67.9 | 59.9 |
| Alghafa: ARC Easy (GK) | 25.0 | 38.1 | 17.5 | 39.2 | 18.9 | 40.9 | 21.2 | **41.1** | **21.5** |
| Okapi: Hellaswag (NLU) | 25.0 | 36.4 | 15.2 | 39.0 | 18.6 | 40.9 | 21.2 | **43.3** | **24.4** |
| OALL2024: PIQA (RES) | 50.0 | 58.4 | 16.8 | 60.9 | 21.8 | 61.8 | 23.6 | **61.9** | **23.8** |
| OALL2024: RACE (RC) | 25.0 | 32.5 | 9.9 | 32.9 | 10.5 | 33.9 | 11.8 | **34.4** | **12.5** |
| OALL2024: SCIQ (GK) | 25.0 | 66.8 | 55.7 | 67.3 | 56.4 | **68.8** | **58.4** | 67.9 | 57.2 |
| X-CODAH (RES) | 25.0 | 35.8 | 14.4 | 34.0 | 12.0 | **40.0** | **19.9** | 38.5 | 18.0 |
| X-CSQA (RES) | 20.0 | 32.8 | 16.0 | 32.5 | 15.6 | 32.7 | 15.8 | **34.2** | **17.8** |
| X-Story Cloze (NLU) | 50.0 | 59.0 | 18.1 | 58.9 | 17.8 | 59.8 | 19.7 | **60.9** | **21.9** |
| ARCD (GK) | 0.0 | 29.9 | 29.9 | 32.0 | 32.0 | 33.0 | 33.0 | **33.1** | **33.1** |
| MLQA (RC) | 0.0 | 22.2 | 22.2 | 21.5 | 21.5 | 21.3 | 21.3 | **23.2** | **23.2** |
| Tydiqa (RC) | 0.0 | **39.5** | **39.5** | 37.9 | 37.9 | 36.8 | 36.8 | 36.5 | 36.5 |
| ArabicMMLU (GK) | 28.0 | 39.9 | 16.6 | 40.0 | 16.7 | 40.1 | 16.9 | **41.1** | **18.2** |
| GK tasks | - | | 27.2 | | 27.6 | | 28.7 | | **29.0** |
| RC tasks | - | | 27.2 | | 26.2 | | 28.0 | | **28.8** |
| RES tasks | - | | 15.7 | | 16.5 | | 19.8 | | **19.8** |
| NLU tasks | - | | 16.6 | | 18.2 | | 20.4 | | **23.1** |
| Aggregate Score | - | | 21.7 | | 22.1 | | 24.2 | | **25.2** |

Table 29: Arabic Results

| Task | Random Baseline | LID | | LID + D | | LID + D + F | | FW2 (R) | |
|---|---|---|---|---|---|---|---|---|---|
| | | Raw | Rescaled | Raw | Rescaled | Raw | Rescaled | Raw | Rescaled |
| Okapi: ARC (GK) | 25.0 | 31.3 | 8.4 | 30.0 | 6.7 | 31.9 | 9.2 | **33.0** | **10.6** |
| Belebele (RC) | 25.0 | 33.8 | 11.7 | 33.1 | 10.8 | 35.2 | 13.5 | **36.0** | **14.6** |
| Okapi: Hellaswag (NLU) | 25.0 | 45.1 | 26.8 | 47.3 | 29.7 | 51.8 | 35.8 | **52.6** | **36.8** |
| X-CODAH (RES) | 25.0 | 37.0 | 15.9 | 37.0 | 16.0 | 40.5 | 20.6 | **42.3** | **23.1** |
| X-CSQA (RES) | 20.0 | 38.0 | 22.5 | 34.6 | 18.2 | 39.8 | 24.7 | **40.4** | **25.5** |
| FQuad (RC) | 0.0 | 28.0 | 28.0 | 27.5 | 27.5 | 29.3 | 29.3 | **35.0** | **35.0** |
| Mintaka (GK) | 0.0 | **9.5** | **9.5** | 7.5 | 7.5 | 8.9 | 8.9 | 8.1 | 8.1 |
| Meta MMLU (GK) | 25.0 | 28.3 | 4.4 | 28.4 | 4.5 | 29.0 | 5.3 | **29.6** | **6.1** |
| GK tasks | - | | 7.4 | | 6.2 | | 7.8 | | **8.3** |
| RC tasks | - | | 19.9 | | 19.2 | | 21.4 | | **24.8** |
| RES tasks | - | | 19.2 | | 17.1 | | 22.7 | | **24.3** |
| NLU tasks | - | | 26.8 | | 29.7 | | 35.8 | | **36.8** |
| Aggregate Score | - | | 18.3 | | 18.0 | | 21.9 | | **23.6** |

Table 30: French Results

| Task | Random Baseline | LID | | LID + D | | LID + D + F | | FW2 (R) | |
|---|---|---|---|---|---|---|---|---|---|
| | | Raw | Rescaled | Raw | Rescaled | Raw | Rescaled | Raw | Rescaled |
| Okapi: ARC (GK) | 25.0 | 29.1 | 5.4 | 30.4 | 7.1 | **33.8** | **11.7** | 32.2 | 9.6 |
| Belebele (RC) | 25.0 | 34.0 | 12.0 | 33.7 | 11.6 | 34.8 | 13.0 | **36.4** | **15.2** |
| Okapi: Hellaswag (NLU) | 25.0 | 41.0 | 21.3 | 43.6 | 24.8 | 45.8 | 27.8 | **46.8** | **29.0** |
| Parus (RES) | 50.0 | 64.9 | 29.9 | 65.7 | 31.4 | **68.2** | **36.4** | 68.1 | 36.2 |
| OpenBookQA (RES) | 25.0 | 36.0 | 14.7 | 36.0 | 14.7 | 35.9 | 14.5 | **38.3** | **17.7** |
| X-CODAH (RES) | 25.0 | 33.9 | 11.8 | 34.9 | 13.2 | 35.4 | 13.8 | **37.1** | **16.2** |
| X-CSQA (RES) | 20.0 | 35.3 | 19.2 | 37.4 | 21.8 | 35.0 | 18.7 | **38.6** | **23.3** |
| X-Story Cloze (NLU) | 50.0 | 66.9 | 33.7 | 66.7 | 33.5 | 68.7 | 37.5 | **69.4** | **38.9** |
| Sber SQuAD (RC) | 0.0 | 27.8 | 27.8 | 32.4 | 32.4 | 32.9 | 32.9 | **37.1** | **37.1** |
| Tydiqa (RC) | 0.0 | 29.9 | 29.9 | 32.4 | 32.4 | **36.7** | **36.7** | 35.5 | 35.5 |
| X-QuAD (RC) | 0.0 | 19.6 | 19.6 | 22.8 | 22.8 | 23.6 | 23.6 | **25.2** | **25.2** |
| RUMMLU (GK) | 25.0 | 29.3 | 5.7 | 29.0 | 5.4 | 29.7 | 6.3 | **30.1** | **6.8** |
| GK tasks | - | | 5.6 | | 6.3 | | **9.0** | | 8.2 |
| RC tasks | - | | 22.3 | | 24.8 | | 26.5 | | **28.2** |
| RES tasks | - | | 18.9 | | 20.2 | | 20.9 | | **23.4** |
| NLU tasks | - | | 27.5 | | 29.1 | | 32.6 | | **34.0** |
| Aggregate Score | - | | 18.6 | | 20.1 | | 22.3 | | **23.4** |

Table 31: Russian Results

| Task | Random Baseline | LID | | LID + D | | LID + D + F | | FW2 (R) | |
|---|---|---|---|---|---|---|---|---|---|
| | | Raw | Rescaled | Raw | Rescaled | Raw | Rescaled | Raw | Rescaled |
| Belebele (RC) | 25.0 | 31.6 | 8.7 | 31.5 | 8.7 | 32.0 | 9.4 | **32.9** | **10.5** |
| Translated Hellaswag (NLU) | 25.0 | 32.5 | 10.0 | 33.1 | 10.8 | 35.9 | 14.5 | **35.9** | **14.5** |
| M3Exam (GK) | 22.9 | 27.6 | 6.1 | **28.1** | **6.7** | 27.5 | 5.9 | 28.1 | 6.7 |
| ThaiQA (RC) | 0.0 | **27.2** | **27.2** | 23.8 | 23.8 | 22.1 | 22.1 | 26.3 | 26.3 |
| X-QuAD (RC) | 0.0 | 19.6 | 19.6 | 18.6 | 18.6 | 17.3 | 17.3 | **20.8** | **20.8** |
| Meta MMLU (GK) | 25.0 | 27.6 | 3.4 | 27.4 | 3.2 | 28.1 | 4.2 | **28.4** | **4.6** |
| GK tasks | - | | 4.7 | | 5.0 | | 5.1 | | **5.6** |
| RC tasks | - | | 18.5 | | 17.0 | | 16.2 | | **19.2** |
| NLU tasks | - | | 10.0 | | 10.8 | | 14.5 | | **14.5** |
| Aggregate Score | - | | 11.1 | | 11.0 | | 11.9 | | **13.1** |

Table 32: Thai Results

| Task | Random Baseline | LID | | LID + D | | LID + D + F | | FW2 (R) | |
|---|---|---|---|---|---|---|---|---|---|
| | | Raw | Rescaled | Raw | Rescaled | Raw | Rescaled | Raw | Rescaled |
| TR Leaderboard: ARC (GK) | 25.0 | 43.7 | 25.0 | 45.1 | 26.8 | **47.9** | **30.6** | 46.3 | 28.4 |
| Belebele (RC) | 25.0 | 31.5 | 8.7 | 32.3 | 9.7 | 33.0 | 10.7 | **34.2** | **12.2** |
| Okapi: Hellaswag (NLU) | 25.0 | 42.4 | 23.3 | 43.3 | 24.3 | 45.3 | 27.1 | **46.8** | **29.1** |
| X-COPA (RES) | 50.0 | 60.7 | 21.3 | 60.7 | 21.5 | **62.8** | **25.6** | 62.7 | 25.3 |
| THQuAD (RC) | 0.0 | 20.4 | 20.4 | 25.6 | 25.6 | 20.6 | 20.6 | **26.1** | **26.1** |
| X-QuAD (RC) | 0.0 | 15.8 | 15.8 | 18.2 | 18.2 | 15.1 | 15.1 | **20.2** | **20.2** |
| Exams (GK) | 23.4 | 29.4 | 7.8 | 29.3 | 7.7 | 28.8 | 7.1 | **30.7** | **9.6** |
| TR Leaderboard: MMLU (GK) | 25.0 | 29.8 | 6.4 | **30.0** | **6.7** | 29.8 | 6.5 | 29.2 | 5.7 |
| GK tasks | - | | 13.1 | | 13.7 | | **14.7** | | 14.6 |
| RC tasks | - | | 15.0 | | 17.8 | | 15.5 | | **19.5** |
| RES tasks | - | | 21.3 | | 21.5 | | **25.6** | | 25.3 |
| NLU tasks | - | | 23.3 | | 24.3 | | 27.1 | | **29.1** |
| Aggregate Score | - | | 18.2 | | 19.3 | | 20.7 | | **22.1** |

Table 33: Turkish Results

| Task | Random Baseline | LID | | LID + D | | LID + D + F | | FW2 (R) | |
|---|---|---|---|---|---|---|---|---|---|
| | | Raw | Rescaled | Raw | Rescaled | Raw | Rescaled | Raw | Rescaled |
| Belebele (RC) | 25.0 | 32.3 | 9.8 | 33.2 | 11.0 | 33.0 | 10.7 | **34.0** | **12.0** |
| C3 (RC) | 27.1 | 47.6 | 28.2 | 47.2 | 27.5 | **50.6** | **32.2** | 49.2 | 30.3 |
| Okapi: Hellaswag (NLU) | 25.0 | 38.3 | 17.7 | 38.6 | 18.1 | 41.4 | 21.9 | **42.2** | **22.9** |
| M3Exam (GK) | 25.9 | 32.8 | 9.3 | 32.6 | 9.0 | 34.1 | 11.1 | **34.3** | **11.3** |
| X-CODAH (RES) | 25.0 | 34.0 | 12.0 | 32.6 | 10.2 | 35.3 | 13.7 | **39.0** | **18.6** |
| X-CSQA (RES) | 20.0 | 38.9 | 23.6 | **41.9** | **27.4** | 41.2 | 26.6 | 39.8 | 24.7 |
| X-COPA (RES) | 50.0 | 60.9 | 21.8 | 62.5 | 25.0 | 62.0 | 24.0 | **64.5** | **28.9** |
| X-Story Cloze (NLU) | 50.0 | 63.0 | 26.0 | 61.6 | 23.1 | 63.1 | 26.2 | **65.5** | **30.9** |
| X-Winograd (NLU) | 50.0 | 70.2 | 40.3 | 70.9 | 41.7 | 72.1 | 44.2 | **74.9** | **49.8** |
| Chinese SQuAD (RC) | 0.0 | 23.5 | 23.5 | 24.1 | 24.1 | 24.1 | 24.1 | **26.3** | **26.3** |
| CMRC (RC) | 0.0 | 38.2 | 38.2 | 38.0 | 38.0 | 38.8 | 38.8 | **40.2** | **40.2** |
| MLQA (RC) | 0.0 | 26.8 | 26.8 | 27.8 | 27.8 | 28.5 | 28.5 | **29.5** | **29.5** |
| AGIEval (ZH subset) (GK) | 26.8 | 32.9 | 8.3 | 33.4 | 9.1 | **34.1** | **10.0** | 33.8 | 9.6 |
| C-Eval (GK) | 25.0 | 31.6 | 8.8 | 32.1 | 9.5 | 32.6 | 10.1 | **32.7** | **10.3** |
| CMMLU (GK) | 25.0 | 32.0 | 9.4 | 33.0 | 10.7 | 34.1 | 12.2 | **34.3** | **12.4** |
| GK tasks | - | | 8.9 | | 9.6 | | 10.9 | | **10.9** |
| RC tasks | - | | 25.3 | | 25.7 | | 26.9 | | **27.7** |
| RES tasks | - | | 19.1 | | 20.9 | | 21.4 | | **24.1** |
| NLU tasks | - | | 28.0 | | 27.7 | | 30.8 | | **34.6** |
| Aggregate Score | - | | 20.3 | | 20.9 | | 22.5 | | **24.3** |

Table 34: Chinese Results

### A.9 Dataset comparison on Canary Languages

In addition to the Reference datasets (Section 3.2, we compare FineWeb2 with the concurrent work de Gibert et al. (2024), as well as with the following language-specific datasets:

- **Arabic**: ArabicWeb24 (Farhat et al., 2024), Arabic-101B (Aloui et al., 2024)
- **French**: Croissant (Faysse et al., 2024)
- **Hindi & Telugu**: Sangraha (Khan et al., 2024)
- **Hindi**: Odaigen (Parida et al., 2024)
- **Russian**: Omnia Russica (Omnia Russica Team, 2024)
- **Thai**: Sea CommonCrawl (Dou et al., 2025)
- **Turkish**: VNGRS-Web-Corpus (Turker et al., 2024)
- **Chinese**: MNBVC (MOP-LIWU Community & MNBVC Team, 2023), Tiger-Bot (TigerResearch, 2023), MAP-CC (Du et al., 2024)

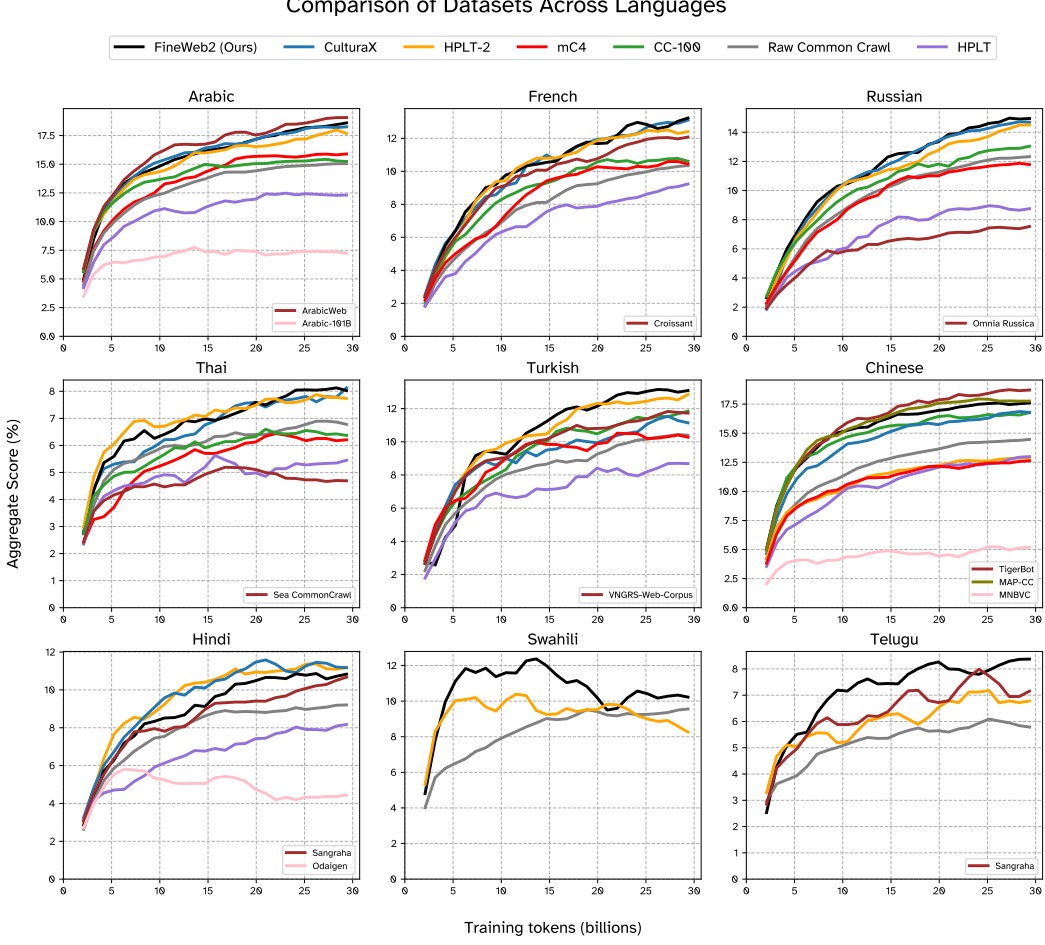

Figure 7: **Per language comparison of FineWeb2** to other multilingual and language-specific datasets. All models were trained for 30 billion tokens. The plots have sliding window smoothing of size 3.

## A.10 Dataset comparison on Unseen Languages

### A.10.1 List of selected evaluation tasks for unseen languages

| Task | Metric | Std |
|---|---|---|
| Meta MMLU (Grattafiori et al., 2024) | Acc (PMI) | 0.0044 |
| Belebele (Bandarkar et al., 2024) | Acc (Token) | 0.0097 |
| Okapi: Hellaswag (Lai et al., 2023) | Acc (Token) | 0.0043 |
| X-CODAH (Lin et al., 2021a) | Acc (Token) | 0.0104 |
| X-CSQA (Lin et al., 2021a) | Acc (Token) | 0.0040 |
| Mintaka (Sen et al., 2022) | F1 | 0.0028 |
| MLQA (Lewis et al., 2020) | F1 | 0.0192 |
| X-QuAD (Artetxe et al., 2020a) | F1 | 0.0134 |

Table 35: Selected tasks for German alongside approximate standard deviation of the scores

| Task | Type | Metric | Std |
|---|---|---|---|
| Okapi: ARC (Lai et al., 2023) | GK | Acc (PMI) | 0.0093 |
| Indo-MMLU (Koto et al., 2023) | GK | Acc (PMI) | 0.0030 |
| Belebele (Bandarkar et al., 2024) | RC | Acc (Token) | 0.0060 |
| Okapi: Hellaswag (Lai et al., 2023) | NLU | Acc (Token) | 0.0063 |
| X-COPA (Ponti et al., 2020) | RES | Acc (Token) | 0.0061 |
| X-Story Cloze (Lin et al., 2021b) | NLU | Acc (Token) | 0.0053 |
| Tydiqa (Clark et al., 2020) | RC | F1 | 0.0120 |

Table 36: Selected tasks for Indonesian alongside approximate standard deviation of the scores

| Task | Type | Metric | Std |
|---|---|---|---|
| Okapi: ARC (Lai et al., 2023) | GK | Acc (PMI) | 0.0119 |
| Meta MMLU (Grattafiori et al., 2024) | GK | Acc (PMI) | 0.0030 |
| X-CSQA (Lin et al., 2021a) | RES | Acc (PMI) | 0.0096 |
| Belebele (Bandarkar et al., 2024) | RC | Acc (Token) | 0.0036 |
| Okapi: Hellaswag (Lai et al., 2023) | NLU | Acc (Token) | 0.0059 |
| M3Exam (Zhang et al., 2023) | GK | Acc (Token) | 0.0038 |
| X-CODAH (Lin et al., 2021a) | RES | Acc (Token) | 0.0203 |
| X-COPA (Ponti et al., 2020) | RES | Acc (Token) | 0.0059 |
| Mintaka (Sen et al., 2022) | GK | F1 | 0.0029 |
| SQuAD-It (Croce et al., 2018) | RC | F1 | 0.0155 |

Table 37: Selected tasks for Italian alongside approximate standard deviation of the scores

| Task | Type | Metric | Std |
|------|------|--------|-----|
| JMMLU (at Waseda University, 2023) | GK | Acc (PMI) | 0.0047 |
| X-CSQA (Lin et al., 2021a) | RES | Acc (PMI) | 0.0168 |
| Belebele (Bandarkar et al., 2024) | RC | Acc (Token) | 0.0047 |
| CommonSenseQA (Kurihara et al., 2022) | RES | Acc (Token) | 0.0089 |
| X-CODAH (Lin et al., 2021a) | RES | Acc (Token) | 0.0088 |
| X-Winograd (Muennighoff et al., 2022) | NLU | Acc (Token) | 0.0092 |
| JSQuAD (Kurihara et al., 2022) | RC | F1 | 0.0117 |

Table 38: Selected tasks for Japanese alongside approximate standard deviation of the scores

| Task | Type | Metric | Std |
|------|------|--------|-----|
| Okapi: ARC (Lai et al., 2023) | GK | Acc (PMI) | 0.0045 |
| Okapi: MMLU (Lai et al., 2023) | GK | Acc (PMI) | 0.0012 |
| X-COPA (Ponti et al., 2020) | RES | Acc (PMI) | 0.0140 |
| Belebele (Bandarkar et al., 2024) | RC | Acc (Token) | 0.0148 |
| Okapi: Hellaswag (Lai et al., 2023) | NLU | Acc (Token) | 0.0099 |
| M3Exam (Zhang et al., 2023) | GK | Acc (Token) | 0.0080 |
| X-CODAH (Lin et al., 2021a) | RES | Acc (Token) | 0.0045 |
| X-CSQA (Lin et al., 2021a) | RES | Acc (Token) | 0.0120 |
| MLQA (Lewis et al., 2020) | RC | F1 | 0.0118 |
| X-QuAD (Artetxe et al., 2020a) | RC | F1 | 0.0067 |

Table 39: Selected tasks for Vietnamese alongside approximate standard deviation of the scores

*A.10.2  Full evaluation results*

| Task | Random Baseline | FineWeb2 (ours) | | Common Crawl | | CulturaX | | HPLT2 | |
|---|---|---|---|---|---|---|---|---|---|
| | | Raw | Rescaled | Raw | Rescaled | Raw | Rescaled | Raw | Rescaled |
| Belebele (RC) | 25.0 | **36.6** | **15.4** | 34.2 | 12.3 | 35.7 | 14.3 | 36.0 | 14.7 |
| Okapi: Hellaswag (NLU) | 25.0 | **42.5** | **23.4** | 37.1 | 16.2 | 40.8 | 21.0 | 41.3 | 21.7 |
| X-CODAH (RES) | 25.0 | 39.7 | 19.6 | 39.1 | 18.8 | **45.0** | **26.7** | 41.8 | 22.4 |
| X-CSQA (RES) | 20.0 | **29.1** | **11.4** | 26.7 | 8.3 | 26.8 | 8.5 | 29.0 | 11.3 |
| Mintaka (GK) | 0.0 | 5.9 | 5.9 | 6.4 | 6.4 | 4.6 | 4.6 | **7.7** | **7.7** |
| MLQA (RC) | 0.0 | 28.1 | 28.1 | 26.2 | 26.2 | 28.7 | 28.7 | **28.9** | **28.9** |
| X-QuAD (RC) | 0.0 | **26.2** | **26.2** | 24.3 | 24.3 | 23.7 | 23.7 | 24.3 | 24.3 |
| Meta MMLU (GK) | 25.0 | 29.5 | 6.0 | 27.9 | 3.8 | 29.0 | 5.3 | **30.0** | **6.7** |
| GK tasks | - | | 6.0 | | 5.1 | | 5.0 | | **7.2** |
| RC tasks | - | | **23.2** | | 20.9 | | 22.2 | | 22.6 |
| RES tasks | - | | 15.5 | | 13.6 | | **17.6** | | 16.8 |
| NLU tasks | - | | **23.4** | | 16.2 | | 21.0 | | 21.7 |
| Aggregate Score | - | | 17.0 | | 13.9 | | 16.4 | | **17.1** |

Table 40: German Results

| Task | Random Baseline | FineWeb2 (ours) | | Common Crawl | | CulturaX | | HPLT2 | |
|---|---|---|---|---|---|---|---|---|---|
| | | Raw | Rescaled | Raw | Rescaled | Raw | Rescaled | Raw | Rescaled |
| Okapi: ARC (GK) | 25.0 | 30.8 | 7.8 | 29.1 | 5.4 | 30.5 | 7.4 | **33.7** | **11.6** |
| Belebele (RC) | 25.0 | 31.8 | 9.1 | 32.0 | 9.3 | **32.3** | **9.7** | 32.1 | 9.5 |
| Okapi: Hellaswag (NLU) | 25.0 | 41.4 | 21.9 | 38.6 | 18.1 | 41.8 | 22.4 | **42.7** | **23.6** |
| X-COPA (RES) | 50.0 | 63.3 | 26.5 | 60.9 | 21.7 | 65.9 | 31.9 | **66.2** | **32.4** |
| X-Story Cloze (NLU) | 50.0 | **66.0** | **32.1** | 63.6 | 27.1 | 63.9 | 27.9 | 65.7 | 31.5 |
| Tydiqa (RC) | 0.0 | 33.6 | 33.6 | **34.6** | **34.6** | 29.0 | 29.0 | 32.3 | 32.3 |
| Indo-MMLU (GK) | 25.0 | 28.9 | 5.2 | 28.7 | 4.9 | 28.0 | 4.0 | **29.6** | **6.1** |
| GK tasks | - | | 6.5 | | 5.1 | | 5.7 | | **8.9** |
| RC tasks | - | | 21.4 | | **21.9** | | 19.4 | | 20.9 |
| RES tasks | - | | 26.5 | | 21.7 | | 31.9 | | **32.4** |
| NLU tasks | - | | 27.0 | | 22.6 | | 25.2 | | **27.6** |
| Aggregate Score | - | | 20.3 | | 17.9 | | 20.5 | | **22.4** |

Table 41: Indonesian Results

| Task | Random Baseline | FineWeb2 (ours) | | Common Crawl | | CulturaX | | HPLT2 | |
|---|---|---|---|---|---|---|---|---|---|
| | | Raw | Rescaled | Raw | Rescaled | Raw | Rescaled | Raw | Rescaled |
| Okapi: ARC (GK) | 25.0 | **32.4** | **9.9** | 28.7 | 4.9 | 30.0 | 6.6 | 30.7 | 7.6 |
| Belebele (RC) | 25.0 | **31.9** | **9.2** | 28.7 | 5.0 | 30.5 | 7.4 | 30.4 | 7.2 |
| Okapi: Hellaswag (NLU) | 25.0 | **45.4** | **27.2** | 38.5 | 18.0 | 43.6 | 24.8 | 44.4 | 25.8 |
| M3Exam (GK) | 33.8 | 39.1 | 8.0 | 38.3 | 6.8 | **40.0** | **9.5** | 38.6 | 7.3 |
| X-CODAH (RES) | 25.0 | **39.3** | **19.1** | 38.7 | 18.2 | 38.0 | 17.3 | 38.7 | 18.2 |
| X-CSQA (RES) | 20.0 | 37.5 | 21.9 | 32.8 | 16.0 | **37.6** | **21.9** | 36.1 | 20.2 |
| X-COPA (RES) | 50.0 | 64.8 | 29.6 | 61.7 | 23.3 | 63.0 | 26.0 | **65.2** | **30.4** |
| Mintaka (GK) | 0.0 | 10.4 | 10.4 | 7.9 | 7.9 | 9.8 | 9.8 | **10.6** | **10.6** |
| SQuAD-It (RC) | 0.0 | 20.3 | 20.3 | 18.2 | 18.2 | **22.2** | **22.2** | 21.8 | 21.8 |
| Meta MMLU (GK) | 25.0 | **30.1** | **6.7** | 29.0 | 5.3 | 29.1 | 5.5 | 29.5 | 5.9 |
| GK tasks | - | | **8.8** | | 6.2 | | 7.8 | | 7.9 |
| RC tasks | - | | 14.7 | | 11.6 | | **14.8** | | 14.5 |
| RES tasks | - | | **23.5** | | 19.2 | | 21.8 | | 22.9 |
| NLU tasks | - | | **27.2** | | 18.0 | | 24.8 | | 25.8 |
| Aggregate Score | - | | **18.6** | | 13.7 | | 17.3 | | 17.8 |

Table 42: Italian Results

| Task | Random Baseline | FineWeb2 (ours) | | Common Crawl | | CulturaX | | HPLT2 | |
|---|---|---|---|---|---|---|---|---|---|
| | | Raw | Rescaled | Raw | Rescaled | Raw | Rescaled | Raw | Rescaled |
| Belebele (RC) | 25.0 | **32.5** | **10.0** | 31.7 | 8.9 | 30.3 | 7.1 | 29.3 | 5.8 |
| CommonSenseQA (RES) | 20.0 | **67.5** | **59.4** | 60.9 | 51.2 | 63.5 | 54.4 | 50.3 | 37.8 |
| X-CODAH (RES) | 25.0 | 37.7 | 16.9 | 37.7 | 16.9 | **38.7** | **18.2** | 37.4 | 16.6 |
| X-CSQA (RES) | 20.0 | 36.4 | 20.5 | 36.4 | 20.5 | **37.2** | **21.5** | 31.0 | 13.7 |
| X-Winograd (NLU) | 50.0 | **60.3** | **20.6** | 54.4 | 8.9 | 57.7 | 15.4 | 59.0 | 18.0 |
| JSQuAD (RC) | 0.0 | **40.5** | **40.5** | 33.1 | 33.1 | 28.5 | 28.5 | 11.7 | 11.7 |
| JMMLU (GK) | 25.0 | **31.7** | **9.0** | 28.9 | 5.1 | 30.7 | 7.5 | 28.7 | 4.9 |
| GK tasks | - | | **9.0** | | 5.1 | | 7.5 | | 4.9 |
| RC tasks | - | | **25.3** | | 21.0 | | 17.8 | | 8.7 |
| RES tasks | - | | **32.2** | | 29.5 | | 31.4 | | 22.7 |
| NLU tasks | - | | **20.6** | | 8.9 | | 15.4 | | 18.0 |
| Aggregate Score | - | | **21.8** | | 16.1 | | 18.0 | | 13.6 |

Table 43: Japanese Results

| Task | Random Baseline | FineWeb2 (ours) | | Common Crawl | | CulturaX | | HPLT2 | |
|---|---|---|---|---|---|---|---|---|---|
| | | Raw | Rescaled | Raw | Rescaled | Raw | Rescaled | Raw | Rescaled |
| Okapi: ARC (GK) | 25.0 | **31.3** | **8.4** | 27.2 | 2.9 | 30.8 | 7.7 | 31.1 | 8.1 |
| Belebele (RC) | 25.0 | 33.0 | 10.6 | 32.6 | 10.2 | 33.1 | 10.8 | **34.1** | **12.1** |
| Okapi: Hellaswag (NLU) | 25.0 | **48.7** | **31.6** | 43.2 | 24.2 | 46.6 | 28.8 | 44.5 | 26.0 |
| M3Exam (GK) | 25.2 | 35.2 | 13.3 | 36.7 | 15.4 | 38.0 | 17.0 | **39.1** | **18.6** |
| X-CODAH (RES) | 25.0 | **40.3** | **20.4** | 35.6 | 14.1 | 38.2 | 17.6 | 38.4 | 17.9 |
| X-CSQA (RES) | 20.0 | 29.6 | 12.0 | 28.5 | 10.6 | **29.8** | **12.3** | 29.7 | 12.2 |
| X-COPA (RES) | 50.0 | **75.7** | **51.3** | 69.7 | 39.5 | 64.6 | 29.2 | 70.5 | 40.9 |
| MLQA (RC) | 0.0 | 19.4 | 19.4 | 18.6 | 18.6 | **23.4** | **23.4** | 22.3 | 22.3 |
| X-QuAD (RC) | 0.0 | 17.3 | 17.3 | 16.9 | 16.9 | **21.3** | **21.3** | 21.2 | 21.2 |
| Okapi: MMLU (GK) | 25.0 | **29.4** | **5.8** | 28.5 | 4.7 | 28.1 | 4.1 | 28.8 | 5.0 |
| GK tasks | - | | 9.2 | | 7.6 | | 9.6 | | **10.6** |
| RC tasks | - | | 15.8 | | 15.3 | | 18.5 | | **18.5** |
| RES tasks | - | | **27.9** | | 21.4 | | 19.7 | | 23.7 |
| NLU tasks | - | | **31.6** | | 24.2 | | 28.8 | | 26.0 |
| Aggregate Score | - | | **21.1** | | 17.1 | | 19.2 | | 19.7 |

Table 44: Vietnamese Results

**A.11 FineWeb2 language composition**

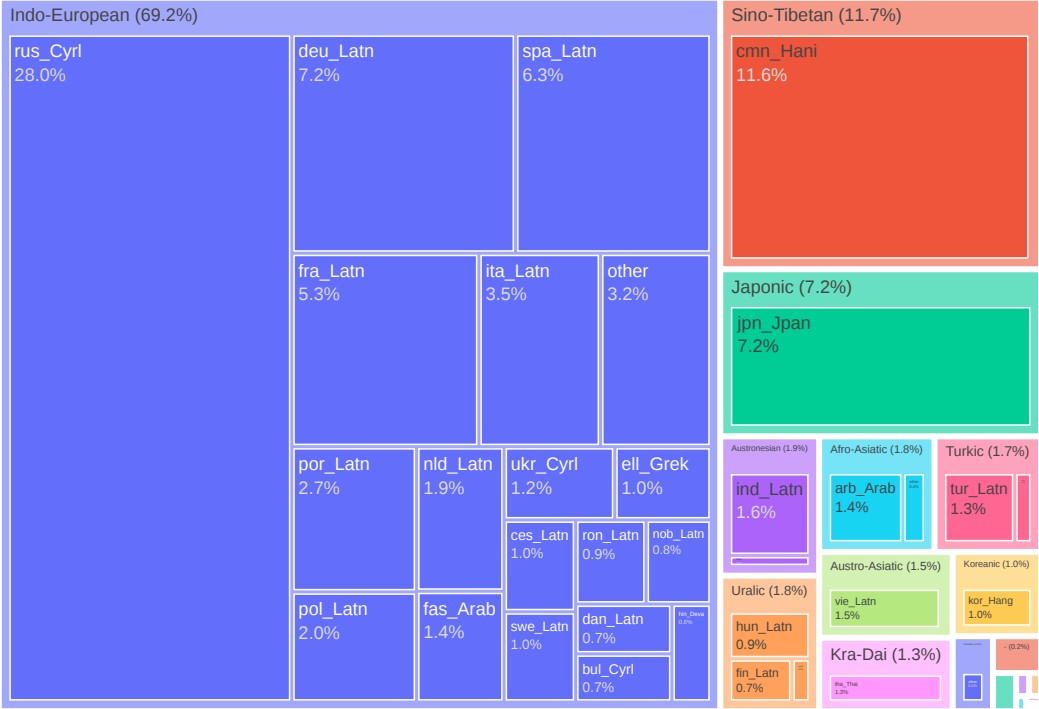

Figure 8: **Language composition of FineWeb2** Distribution of languages in the final FineWeb2 dataset. Percentages refer to total utf-8 bytes of each language or language family.

Table 45: FineWeb2 80 largest language stats

| ISO 639-3 | Script | Name | Language Family | Words | Documents | Disk size |
|---|---|---|---|---|---|---|
| rus | Cyrl | Russian | Indo-European | 588,579,493,780 | 699,083,579 | 5.82TB |
| cmn | Hani | Mandarin Chinese | Sino-Tibetan | 543,543,038,750 | 636,058,984 | 2.42TB |
| deu | Latn | German | Indo-European | 262,271,052,199 | 495,964,485 | 1.51TB |
| jpn | Jpan | Japanese | Japonic | 331,144,301,801 | 400,138,563 | 1.50TB |
| spa | Latn | Spanish | Indo-European | 261,523,749,595 | 441,287,261 | 1.32TB |
| fra | Latn | French | Indo-European | 220,662,584,640 | 360,058,973 | 1.11TB |
| ita | Latn | Italian | Indo-European | 139,116,026,491 | 238,984,437 | 739.24GB |
| por | Latn | Portuguese | Indo-European | 109,536,087,117 | 199,737,979 | 569.24GB |
| pol | Latn | Polish | Indo-European | 73,119,437,217 | 151,966,724 | 432.01GB |
| nld | Latn | Dutch | Indo-European | 74,634,633,118 | 147,301,270 | 397.51GB |
| ind | Latn | Indonesian | Austronesian | 60,264,322,142 | 100,238,529 | 348.65GB |
| vie | Latn | Vietnamese | Austro-Asiatic | 50,886,874,358 | 61,064,248 | 319.83GB |
| fas | Arab | Persian | Indo-European | 39,705,799,658 | 58,843,652 | 304.62GB |
| arb | Arab | Standard Arabic | Afro-Asiatic | 32,812,858,120 | 61,977,525 | 293.59GB |
| tur | Latn | Turkish | Turkic | 41,933,799,420 | 95,129,129 | 284.52GB |
| tha | Thai | Thai | Kra-Dai | 24,662,748,945 | 35,897,202 | 278.68GB |
| ukr | Cyrl | Ukrainian | Indo-European | 25,586,457,655 | 53,101,726 | 254.86GB |
| ell | Grek | Modern Greek (1453-) | Indo-European | 22,827,957,288 | 47,421,073 | 222.05GB |
| kor | Hang | Korean | Koreanic | 48,613,120,582 | 60,874,355 | 213.43GB |
| ces | Latn | Czech | Indo-European | 35,479,428,809 | 66,067,904 | 206.33GB |
| swe | Latn | Swedish | Indo-European | 35,745,969,364 | 59,485,306 | 202.96GB |
| hun | Latn | Hungarian | Uralic | 30,919,839,164 | 49,935,986 | 199.69GB |
| ron | Latn | Romanian | Indo-European | 35,017,893,659 | 58,303,671 | 186.19GB |
| nob | Latn | Norwegian Bokmål | Indo-European | 32,008,904,934 | 38,144,343 | 172.05GB |
| dan | Latn | Danish | Indo-European | 28,055,948,840 | 45,391,655 | 150.72GB |
| bul | Cyrl | Bulgarian | Indo-European | 16,074,326,712 | 25,994,731 | 145.75GB |
| fin | Latn | Finnish | Uralic | 20,343,096,672 | 36,710,816 | 143.03GB |
| hin | Deva | Hindi | Indo-European | 11,173,681,651 | 22,095,985 | 120.98GB |
| ben | Beng | Bengali | Indo-European | 6,153,579,265 | 15,185,742 | 87.04GB |
| slk | Latn | Slovak | Indo-European | 14,808,010,769 | 29,991,521 | 85.43GB |
| heb | Hebr | Hebrew | Afro-Asiatic | 8,462,976,117 | 14,491,748 | 68.71GB |

Table 45 – Continued from previous page

| ISO 639-3 | Script | Name | Language Family | Words | Documents | Disk size |
|---|---|---|---|---|---|---|
| lit | Latn | Lithuanian | Indo-European | 9,132,828,961 | 13,471,965 | 56.50GB |
| bos | Latn | Bosnian | Indo-European | 9,086,837,979 | 21,243,255 | 49.18GB |
| slv | Latn | Slovenian | Indo-European | 7,688,373,264 | 12,059,130 | 41.80GB |
| ekk | Latn | Standard Estonian | Uralic | 6,564,292,000 | 10,218,587 | 40.82GB |
| cat | Latn | Catalan | Indo-European | 8,348,091,726 | 17,136,414 | 40.35GB |
| tam | Taml | Tamil | Dravidian | 1,937,150,898 | 5,528,854 | 36.97GB |
| hrv | Latn | Croatian | Indo-European | 6,609,299,440 | 6,195,824 | 35.91GB |
| lvs | Latn | Standard Latvian | Indo-European | 5,371,151,279 | 8,030,316 | 33.36GB |
| zsm | Latn | Standard Malay | Austronesian | 5,648,387,840 | 9,421,248 | 31.94GB |
| azj | Latn | North Azerbaijani | Turkic | 3,894,255,826 | 7,291,231 | 26.90GB |
| srp | Cyrl | Serbian | Indo-European | 2,858,500,314 | 4,146,124 | 26.87GB |
| kat | Geor | Georgian | Kartvelian | 1,439,572,993 | 3,706,659 | 25.23GB |
| npi | Deva | Nepali (individual language) | Indo-European | 1,642,856,349 | 4,888,163 | 25.13GB |
| mar | Deva | Marathi | Indo-European | 1,541,225,070 | 3,912,702 | 22.57GB |
| mal | Mlym | Malayalam | Dravidian | 1,054,187,581 | 3,322,526 | 22.27GB |
| kaz | Cyrl | Kazakh | Turkic | 1,876,843,453 | 3,344,366 | 20.67GB |
| urd | Arab | Urdu | Indo-European | 2,733,266,493 | 4,809,542 | 19.93GB |
| als | Latn | Tosk Albanian | Indo-European | 3,454,387,059 | 8,597,826 | 18.18GB |
| mkd | Cyrl | Macedonian | Indo-European | 1,611,392,841 | 4,150,902 | 14.99GB |
| tel | Telu | Telugu | Dravidian | 891,002,487 | 1,964,395 | 14.42GB |
| kan | Knda | Kannada | Dravidian | 748,850,327 | 2,390,982 | 12.91GB |
| mya | Mymr | Burmese | Sino-Tibetan | 854,400,671 | 1,558,304 | 12.35GB |
| guj | Gujr | Gujarati | Indo-European | 934,124,052 | 2,127,094 | 11.71GB |
| bel | Cyrl | Belarusian | Indo-European | 1,166,541,148 | 2,100,873 | 11.47GB |
| isl | Latn | Icelandic | Indo-European | 1,696,354,360 | 3,014,429 | 10.27GB |
| khm | Khmr | Khmer | Austro-Asiatic | 667,495,692 | 1,586,460 | 8.70GB |
| khk | Cyrl | Halh Mongolian | Mongolic | 824,211,882 | 1,622,882 | 8.52GB |
| fil | Latn | Filipino | Austronesian | 1,636,238,017 | 2,349,050 | 8.13GB |
| ary | Arab | Moroccan Arabic | Afro-Asiatic | 843,523,994 | 2,365,405 | 7.74GB |
| afr | Latn | Afrikaans | Indo-European | 1,598,352,868 | 1,992,040 | 7.69GB |
| hye | Armn | Armenian | Indo-European | 634,273,060 | 1,757,415 | 7.17GB |
| sin | Sinh | Sinhala | Indo-European | 512,453,069 | 1,185,323 | 7.05GB |
| glg | Latn | Galician | Indo-European | 1,236,233,473 | 2,522,814 | 6.47GB |
| uzn | Cyrl | Northern Uzbek | Turkic | 544,866,919 | 1,357,811 | 6.12GB |

Table 45 – Continued from previous page

| ISO 639-3 | Script | Name | Language Family | Words | Documents | Disk size |
|---|---|---|---|---|---|---|
| pan | Guru | Panjabi | Indo-European | 522,788,467 | 944,160 | 5.64GB |
| ory | Orya | Odia | Indo-European | 333,760,951 | 1,298,188 | 4.92GB |
| uzn | Latn | Northern Uzbek | Turkic | 687,002,994 | 1,233,463 | 4.45GB |
| kir | Cyrl | Kirghiz | Turkic | 397,449,282 | 1,069,582 | 4.36GB |
| eus | Latn | Basque | Language isolate | 711,939,889 | 1,569,434 | 4.30GB |
| lat | Latn | Latin | Indo-European | 714,764,848 | 1,473,541 | 3.86GB |
| tgk | Cyrl | Tajik | Indo-European | 396,209,383 | 688,384 | 3.75GB |
| gmh | Latn | Middle High German (ca. 1050-1500) | Indo-European | 506,396,917 | 84,495 | 3.41GB |
| swh | Latn | Swahili (individual language) | Niger-Congo | 569,542,024 | 1,206,300 | 3.08GB |
| arz | Arab | Egyptian Arabic | Afro-Asiatic | 345,040,810 | 853,290 | 2.92GB |
| nno | Latn | Norwegian Nynorsk | Indo-European | 522,740,774 | 1,214,870 | 2.68GB |
| cym | Latn | Welsh | Indo-European | 523,226,616 | 831,878 | 2.50GB |
| amh | Ethi | Amharic | Afro-Asiatic | 239,936,286 | 428,373 | 2.49GB |
| pbt | Arab | Southern Pashto | Indo-European | 337,138,269 | 639,983 | 2.41GB |
| ckb | Arab | Central Kurdish | Indo-European | 236,342,609 | 554,993 | 2.39GB |
| ...[7] | | | | | | |
| **Total** | | | | **3,339,271,691,958** | **5,018,505,566** | **20.78TB** |

---

[7]Full list available at https://github.com/huggingface/fineweb-2/blob/main/fineweb2-language-distribution.csv

### A.12 Bible and Wikipedia content

For each language low resource language, we first compiled the distribution of documents by domain name. We then averaged the frequency of each domain across all languages, to find specific domains that were a common source of data for different languages (which from manual inspection was the case for specific Bible websites and Wikipedia). We manually labeled the top domains that belonged to Bible or Wikipedia websites (Table 46), and then measured the fraction of each language corpora that belonged to these domains. Out of 1868 language-script pairs in the final dataset, 70% (1320 of them) have more than half their documents from Bible- or Wikipedia-related domains. This is mostly driven by Bible content, as can be seen in Fig. 9.

| Bible Domains | Wiki Domains |
|---|---|
| ebible.org | wikipedia.org |
| bible.is | wikimedia.org |
| jw.org | wikisource.org |
| stepbible.org | wiktionary.org |
| bibles.org | |
| bible.com | |
| breakeveryyoke.com | |
| png.bible | |
| americanbible.org | |
| pngscriptures.org | |
| globalrecordings.net | |
| gospelgo.com | |
| httlvn.org | |
| biblegateway.com | |
| jesusforafrica.net | |
| bible.com.au | |
| pacificbibles.org | |
| scriptureearth.org | |
| divinerevelations.info | |
| beblia.com | |
| aboriginalbibles.org.au | |
| eevangelize.com | |
| biblica.com | |
| e-alkitab.org | |
| alkitab.pw | |
| amazinggracebibleinstitute.com | |
| bibleforchildren.org | |
| aionianbible.org | |
| cyber.bible | |
| biblehub.com | |
| myanmarbs.org | |
| baebol.org | |
| christianchildmultilingualbibleverse.wordpress.com | |
| femissionaria.blogspot.com | |
| biblics.com | |
| churchofjesuschrist.org | |
| biblesa.co.za | |
| bible-tools.org | |
| torresstraitbibles.org.au | |

Table 46: List of Bible-related and Wiki-related domains

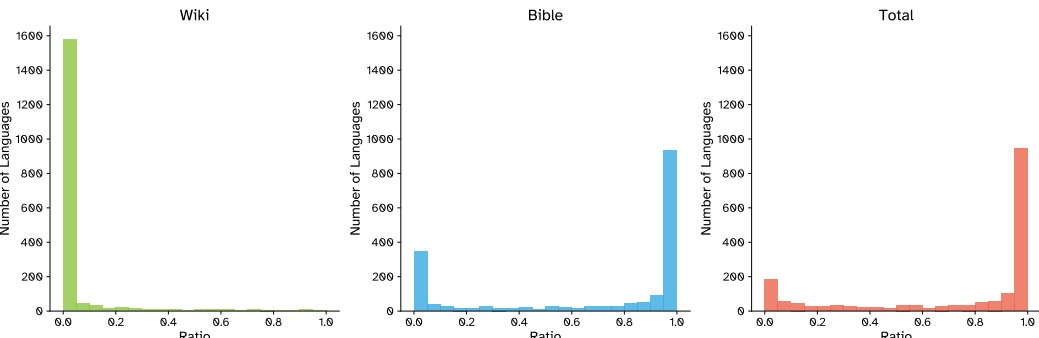

Figure 9: **Ratio of Wikipedia and Bible content per language** Most languages have a small fraction of their content originating from Wikipedia (with some exceptions). Bible content, on the other hand, is a big part of the corpora of many lower-resource languages.

### A.13 Train-Test Split

Our dataset release is split into a train and test set, per language. The test set should not be used for training but instead can help research questions such as on memorization or data attribution. The test set is obtained as a random subset (by a hash function applied on the document content), and contains $\min\{1\%, 100k\}$ of the documents per language pre-filtering, with a reduction in size when these documents are filtered with the same process as the train set. It is only provided for languages of sufficient size.

