# OpenReview forum: "FineWeb2: One Pipeline to Scale Them All — Adapting Pre-Training Data Processing to Every Language"
_colmweb.org/COLM/2025/Conference — COLM 2025_

### Official Review · Reviewer_5fxD · 2025-04-19

**Rating:** 9
**Confidence:** 4
**Ethics Flag:** 1

**Summary:**

The paper introduces FineWeb2, a multilingual dataset spanning over 1000 languages. Unlike previous multilingual datasets that adopt one-size-fits-all approaches, FineWeb2 introduces per-language pipelines informed by language-specific statistics. The design choices in the pipelines are guided by the evaluation of monolingual models trained on nine representative languages. To select the evaluation datasets, the paper also proposes a set of criteria for identifying tasks that provide early quality signals. The paper then compares monolingual models trained on FineWeb2 with ones trained on alternative multilingual datasets, both on seen and unseen languages.

**Questions To Authors:**

How did you choose which documents to keep after deduplication? Since Zyda2 [1] reports different quality levels for documents within the same cluster, did you investigate the impact of your selection?

[1] https://www.zyphra.com/post/building-zyda-2#zyda2_5

**Reasons To Accept:**

The paper releases a large multilingual dataset covering an extensive set of languages, together with the code for the data pipeline and remaining experiments, which I believe will be valuable for the community working on language modeling.

The development process of FineWeb2 is also extensively detailed, including comprehensive ablations to justify the choices made which validate the quality of the proposed approach. Additionally, the methodology for selecting evaluation datasets providing early quality signals is interesting, and may be used in future work curating vast amounts of data.

Finally, the finding that data with either no repetition or very high repetition is of lower quality is intriguing. Also, showing that rehydrating the dataset based on the filtering applied within a deduplication cluster is effective is interesting.

**Reasons To Reject:**

Besides training monolingual datasets, a multilingual dataset will likely also be used for training multilingual models. Therefore, I believe it would strengthen the paper to compare a multilingual model on FineWeb2 with those trained on existing alternatives.

---

> ### Author Response · Authors · 2025-06-01
>
> We thank reviewer `5fxD` for the very positive assessment of our work and for highlighting the deduplication and rehydration parts of the paper.
>
> > How did you choose which documents to keep after deduplication? Since Zyda2 [1] reports different quality levels for documents within the same cluster, did you investigate the impact of your selection?
>
> We thank the reviewer for their thoughtful question regarding deduplication. While in this work we simply chose a random document to keep for each cluster, we plan to further explore data diversity within duplicate clusters in future work. We believe that a simple change to our rehydration pipeline could provide some gains: instead of simply repeating a document N times, we could instead sample N distinct (and potentially while maximizing their diversity) documents from each cluster.
>
> The Zyda2 findings are quite interesting but we would like to point out that in our understanding, the specific result regarding the drastically different classifier scores for extremely similar documents (with only a handful of words changed) suggest a limitation of the classifiers themselves, and not that the same document without “More You may also like…” produces a significantly stronger model.
>
> > Besides training monolingual datasets, a multilingual dataset will likely also be used for training multilingual models. Therefore, I believe it would strengthen the paper to compare a multilingual model on FineWeb2 with those trained on existing alternatives.
>
> While we agree in principle, a multilingual model introduces additional confounders such as the proportion of data from each language (mixing), as well as cross-language transferability (we would not be sure that observed gains in a language would be exclusively due to the improved quality of that language’s data). Additionally, we would still be limited by scalability issues in terms of evaluation tasks: besides translation tasks which are generally available, we’d still need to curate benchmarks for other tasks for each additional language.

---

> > ### Comment · Reviewer_5fxD · 2025-06-03
> >
> > Thank you for your answers; they have addressed my concerns.
> >
> > I have no further questions for the authors during the remainder of the rebuttal period, and I would like to thank them again for their clarifications.

---

### Official Review · Reviewer_faKg · 2025-04-24

**Rating:** 8
**Confidence:** 3
**Ethics Flag:** 1

**Summary:**

This paper describes a pre-processing pipeline for large-scale, multilingual datasets that are built on data scraped from the web. Instead of applying the same cleaning and filtering across all languages, the authors propose to tailor the specific algorithms and their parameters to the individual languages. Furthermore, they provide a large number of ablations on a selected set of 9 languages to evaluate the effect of individual cleaning/filtering steps.

Overall, the amount of work behind this paper is quite impressive with the sheer number of ablation runs. The resulting corpus covers a large variety of languages and scripts and seems to hold up very well when used as training data for downstream tasks, at least for the evaluated 14 languages (out of 1000).

The only reservation I have about the paper in its current form is that it is much too detailed/long. The regular 9 pages is all text, while all but one of the figures/tables have been pushed to the appendix (which is >80 (!) pages).

**Questions To Authors:**

- casing: when referencing figures and tables: uppercase (*Fig. 3*, *Table 6*). Same with references to the appendix (*see Appendix A.5*)
- for the ablation experiments with the 9 canary languages: why use the Gemma tokenizer instead of just training language specifc ones (with a smaller vocab)? I don't think it makes much sense to use a 250k vocab to train monolingual models (would also save some computation for the ablations)
- line 320: average of ranks - what exactly are the ranks?
- line 402: trained **on** 29 billion..
- line 406-407: prior multilingual dataset**s** (plural)
- line 412: These trends holds up.. sg/pl

**Reasons To Accept:**

The paper is generally well written, clear and easy to follow, and the resulting corpus is certainly of interest for the community. The individual processing steps are evaluated with an impressive amount of ablation experiments.

Intuitively, the idea to tailor settings to the specific languages instead of using a one-size-fits-all approach makes a lot of sense, and the provided evaluations on downstream tasks seem to confirm this.

**Reasons To Reject:**

As mentioned above, in its current form, the paper is too detailed. It fits the 9 page limit only because figures/tables have been moved to the appendix. The reader has to flip back and forth between text and appendix.
I would strongly recommend to condense the main part, maybe even move some of the individual steps entirely to the appendix, and instead include the most relevant figures/tables in the main text.

Something else to point out, although this is relatively minor: There is a tail end of very small minority languages that have very little data (<5 documents). I'm a bit suspicious regarding the quality of that data, as previous work (e.g. Kreutzer et al. 2022) has shown that in those cases, data quality is often quite bad (different language or not language data at all, simple phrases instead of fluent text, etc.). It might be worth to verify those (at least check if the language was correctly identified), and if necessary remove them.

---

> ### Author Response · Authors · 2025-06-01
>
> We would like to thank reviewer `faKG` for their detailed review and for raising interesting points regarding the tail end of minority languages. We address their concerns below.
>
> ## Appendix
>
> We agree with the reviewer’s concerns regarding the use of the Appendix and will update our draft to better integrate some of the results the reviewer mentions.
>
> ## Tail end of minority languages
>
> We believe the reviewer raises a very relevant point regarding languages that have no (or extremely low quality) content. We share the reviewer’s concerns regarding the quality of languages with very few documents in the final dataset, which is what motivated us to develop the “Precision filtering lower resource languages” section, inspired by work related to the reference the reviewer mentions (Caswell et al. (2020); Bapna et al. (2022)).
> The version of the table with language composition present in the draft is from before we applied this final step, hence why those tail end languages are shown with such short document counts. We will update the paper with the final data and likely replace the table with plots showing the composition, in line with the reviewer’s criticism of the excessive number of appendix pages, half of which are taken up by this final table.
>
> ## Tokenizer and evaluation
>
> > for the ablation experiments with the 9 canary languages: why use the Gemma tokenizer instead of just training language-specific ones (with a smaller vocab)? I don't think it makes much sense to use a 250k vocab to train monolingual models (would also save some computation for the ablations)
>
> Not training our own tokenizer (usually on our own data) ensures that we do not unfairly benefit our own dataset. Since most languages do not have state-of-the-art small vocabulary tokenizers available, we opted instead to choose from among the well-known multilingual vocabularies with a large vocabulary size. Furthermore, though we agree with the reviewer regarding the additional cost, using the same tokenizer for all languages allows us to simplify our codebase and avoid accidental mistakes.
>
> > line 320: average of ranks - what exactly are the ranks?
>
> We will clarify this part. We compute the aggregate score of each method for each language (as described in the evaluation section) and then sort the methods by this score (rank 1 means a method achieved the highest aggregate score in a given language, etc) and average the ranks across languages.
>
> ## Typography
>
> We thank the reviewer for pointing out typographical errors and will correct them.

---

### Official Review · Reviewer_RcCQ · 2025-05-11

**Rating:** 8
**Confidence:** 4
**Ethics Flag:** 2

**Summary:**

The paper presents a new dataset, FineWeb2, a 19T corpus covering over 1000 languages created with a specialized pipeline. The authors ablate the preprocessing decisions (langid, deduplication strategy, filtering recipe) on 9 languages covering different scripts and resourcedness using evaluation tasks carefully chosen to get a meaningful signal during training. They compare the collected dataset against existing pipelines and show that their proposed baseline outperforms one-size-fits-all solutions typically used for creating multilingual pre-training datasets. However, room for improvement exists relative to language-specialized baselines.

**Ethics Concerns Details:**

While there might not be a direct risk but since there are model released on large quantity of web-crawled data, it would be useful to include a potential risk statement at the end of the paper.

**Questions To Authors:**

1. Non-random performance early on: L159-162: Does this argument conflate task difficulty with task usefulness? The lack of signal from ablations at a specific model scale could be attributed to limited model capacity, rather than indicating that the task itself is not useful.
2. While the authors present results on "unseen" languages, given that the tasks are selected based on the model learning signal itself,  there is a risk of circular reasoning.
3. L260-262: could it be because the deduplication was done globally per language? Was there any analysis on cross-language "information" overlap?

**Reasons To Accept:**

1. The resources released with this paper and the findings will significantly help the development of better multilingual models and research in understanding the impact of data quality on the trained models.
2. The experiments and ablations in the paper are thorough and principally designed.

**Reasons To Reject:**

While this is a very strong paper, it would have been helpful to present the main results within the main body of the paper. Currently, all tables and figures, except for those from Paper 1, are placed in the appendix. Given the scale of the experiments, it’s understandable that space constraints are a challenge, but the main paper should ideally be self-contained. I believe the authors could have made a greater effort to integrate key results into the main paper. Relatedly, the proposed selection of evaluation tasks is vaguely defined without giving enough background on why these choices were necessary.

---

> ### Author Response · Authors · 2025-06-01
>
> We would like to thank reviewer `RcCQ` for their positive assessment of our work, and specifically for recognizing the importance and potential impact of our released resources and findings. We appreciate the relevant questions raised by the reviewer regarding evaluation and address their concerns below.
>
> ## Appendix
>
> We agree with the reviewer’s concerns regarding the use of the Appendix and will update our draft to better integrate some of the results the reviewer mentions.
>
> ## Evaluation tasks
>
> > Relatedly, the proposed selection of evaluation tasks is vaguely defined without giving enough background on why these choices were necessary.
>
> Is the reviewer referring to the final list of evaluation tasks (presented in Appendix A.5.3) or to the criteria used to select tasks (mentioned in Section 3.3 and formally defined in Appendix A.5)?
>
> In the first paragraph of Section 3.3 we justify the need to select tasks ourselves, since for most languages no “standard” evaluation suites are available, and many tasks have subpar quality (originating from machine translation).
>
> We would be happy to update Section 3.3 if the reviewer is willing to clarify which part of the evaluation setup they believe requires further background. Additionally, we can prioritize moving information from Appendix A.5 into the main text to help clarify.
>
> > Non-random performance early on: L159-162: Does this argument conflate task difficulty with task usefulness? The lack of signal from ablations at a specific model scale could be attributed to limited model capacity, rather than indicating that the task itself is not useful.
>
> The reviewer raises a great point that we are happy to clarify. We consider a task to be useful if it allows us to make informed comparisons between different datasets/models at the scale we run our experiments. This means that a task that provides non-random performance at a large scale but not at our small scale (due to its difficulty) is not useful for our experiments (even if it could be at a larger scale). We explicitly mention this as a limitation in the Conclusion: “[...] we studied “high-signal” properties of each task at the very early stages of model training, and so it is possible that the properties could change significantly as training progresses, making some tasks more viable.”
>
> > While the authors present results on "unseen" languages, given that the tasks are selected based on the model learning signal itself, there is a risk of circular reasoning.
>
> This is a reasonable concern that we were not able to avoid due to the previously mentioned lack of established benchmarks for most languages. Out of a concern that our task selection could unfairly benefit our own dataset, we selected tasks using a similar approach to the canary languages: we relied on other datasets (excluding our own) to quantify task-signal.
>
> ## Deduplication
>
> > L260-262: could it be because the deduplication was done globally per language? Was there any analysis on cross-language "information" overlap?
>
> While we did not perform the mentioned analysis, we investigated a related question: Is the different impact per-language related to the degree of duplication in each language? We looked at the percentage of data removed during deduplication for each language, but the languages sorted by this value (see table below) do not align with the order of performance impact: French and Chinese, two languages that both show little benefit from deduplication, are ranked on opposite ends of the table. As both the language with the most (removed) repeated content and the one with the least showed very similar performance effects after deduplication, it seems that the degree of duplication is not the leading factor in the disparate results we observe across languages. We also considered other hypotheses, but were similarly unable to draw any definitive conclusions.
>
> | Code | Script | Name                          | Removal % |
> |------|--------|-------------------------------|-----------|
> | fra  | Latn   | French                        | 83.03%    |
> | tur  | Latn   | Turkish                       | 83.02%    |
> | swh  | Latn   | Swahili (individual language) | 80.03%    |
> | rus  | Cyrl   | Russian                       | 79.83%    |
> | arb  | Arab   | Standard Arabic               | 76.83%    |
> | tha  | Thai   | Thai                          | 75.26%    |
> | tel  | Telu   | Telugu                        | 71.87%    |
> | arb  | Latn   | Standard Arabic               | 71.29%    |
> | hin  | Deva   | Hindi                         | 69.52%    |
> | cmn  | Hani   | Mandarin Chinese              | 68.69%    |

---

> > ### Comment · Reviewer_RcCQ · 2025-06-03
> > **Response to rebuttal**
> >
> > Thanks for the response and sharing additional insights on the deduplication.
> >
> > I meant the criteria used to select the evaluation tasks as it is unclear if the criteria themselves could inject any bias and whether the expectation is to use such criteria at the specific ablated scale. This is clarified in the response for the question  "..Non-random performance early on: L159-162:" I apologize I missed the note in the conclusion. I believe it would be useful to discuss this in the main body or even as a footnote next to the key criteria as opposed to the conclusion because they ground the usefulness of the metrics used for selection for use cases (e.g. model sizes).

---

> ### Author Response · Authors · 2025-06-03
>
> >  believe it would be useful to discuss this in the main body or even as a footnote next to the key criteria as opposed to the conclusion because they ground the usefulness of the metrics used for selection for use cases (e.g. model sizes).
>
> Indeed we should have made it clearer when we first introduce the task selection process that it targets the specific scale at which we run our ablations, we will update our draft to highlight this aspect.

---

### Official Review · Reviewer_A835 · 2025-05-12

**Rating:** 8
**Confidence:** 5
**Ethics Flag:** 1

**Summary:**

The submission presents the FineWeb2 dataset, a new 19 terabyte (4.5 billion documents) multilingual dataset covering more than 1000 languages created using an adaptable processing pipeline also described in the paper; the dataset is based on almost 100 Common Crawl dumps. These are the paper’s two important and highly useful contributions in addition to showing that the pipeline can be used to create non-English datasets that, in turn, can be used to develop models that are better than models produced with other datasets.

Originality: this submission is very hands-on, it presents incremental work which is based on the original FineWeb and other results from the community that are used to enhance the approach. The more original or innovative solutions presented in the paper all relate to smaller improvements that tackle clear and practical issues when developing a pipeline meant for creating pre-training datasets in the range of terabytes. The fact that the processing pipeline can be automatically (or almost automatically) adapted to a certain language is an original idea (though it’s one that many colleagues have pondered in the past).

Significance: despite the rather low originality, the paper’s significance is very high since it presents an important dataset covering more than a 1000 languages, carefully produced paying attention to overall data quality.

Quality and Clarity: while the overall quality of the paper is, more or less, high, one specific aspect must be pointed out. The submitted PDF file has 105 (!) pages. It includes the nine pages of the main body of the paper, 14 pages of references and 82 pages of appendices. This length is, for the lack of a better term, excessive. In many parts throughout the main part of the paper (i.e., the first nine pages), the authors refer to tables or figures in the appendix. In several of these cases, this reviewer considers the tables or figures included in the appendix as critical to the overall understanding of the paper. While other colleagues may see this differently, this reviewer considers this practice appendix mis-use. This reviewer would expect a more careful curation of the content, putting all the tables and figures required for understanding the main part of the paper in the actual main part of the paper. This relates to, among others, the lists of tasks and metrics (A.5.3), Figure 3, Figure 4, Figure 7, Figure 8, Appendix A.8.3 (among others). Closely related to the previous point, my second main critical remark relates to several sections in the paper that feel under-explored and that should’ve received more attention when drafting the paper: confidence thresholds (Section 4.2, last paragraph), deduplication (Section 4.3, last paragraph, impact on data quality), filtering recipes (Section 4.4, statistics on each language on different corpora, different metrics etc.), stopwords (Section 4.4.1, why at least two words from the stopwords list?), precision filtering (Section 4.4.3). This reviewer does understand that space is limited in every paper but in this specific case, the submission is a bit unbalanced in terms of level of details provided in the main body of the paper.

A few more detailed comments:

- The paper talks about “early-signal” benchmarks and “high-signal” benchmarks. Are they the same? Is there a difference?
- The authors use the term “canary languages”, i.e., the languages they’re testing their work with (akin to canary birds in a coal mine, I assume). While I understand the metaphor, some colleagues probably won’t so maybe call them “test languages”.
- Instead of footnote 3 pointing to a Wikipedia page, a reference from linguistics research would be more appropriate.
- Page 3 – improve coherence: “Unfortunately, all datasets except raw CommonCrawl contained a limited amount of data for Telugu and Swahili, …” => “Unfortunately, all datasets except raw CommonCrawl only contained a rather limited amount of data for Telugu and Swahili, …”
- Section 4.1: the mentioned “blocklist” was not questioned at all but simply accepted as fit for purpose?
- Last sentence on page 4: as FineWeb covers English only and FineWeb2 covers everything except English, I think it would make a lot of sense to state explicitly that FineWeb2 *complements* FineWeb or that it provides data and language coverage that is *complementary* to FineWeb.
- Several times phrases like “the concurrent work HPLTv2.0” are mentioned. Why exactly is HPLTv2.0 singled out and labelled as “concurrent work”?
- The choice for GlotLID (paragraph 3 on page 5) needs a more comprehensive explanation.
- Section 4.4.3: I would’ve expected a discussion of mixed-languages documents here (or somewhere else in this paper).
- The citation “(Consortium, 2025)” should probably be “(PLLuM Consortium, 2025)”.
- Line 256: “the number *of* documents”
- Page 6: “wikipedia” vs. “Wikipedia”
- Line 387: the word “preexisting” in “preexisting non-English datasets” is weird. Perhaps use “other non-English datasets” or simply “non-English datasets”.
- Line 405: “are shown” -> “is shown”.
- Line 427: “as wide possible” -> “as wide as possible”
- Sometimes wrong quotation marks are used (sorry typography nerd here).
- When referring to specific tables, figures or appendices, it’s “Figure X”, “Table X” and “Appendix X” (with a capital letter at the beginning of the word).
- All numbers until ten should be written out as words (3 -> three, 9 -> nine).

**Questions To Authors:**

See above under "Summary".

**Reasons To Accept:**

- Novel dataset, based on Common Crawl, quality filtered, covering 1000+ languages. First very important artefact!
- Improved CC processing pipeline with automated parts that automatically adapt the pipeline to a certain language. Second very important artefact!
- Many interesting ablation experiments that clearly demonstrate the value and usefulness of the work presented in the paper.
- All code, data etc. is made available.

**Reasons To Reject:**

- Appendix mis-use – many critical and important parts are buried in the appendix and linked to from the main body of the paper (see above).
- Certain parts of the paper are under-explored (see above).

---

> ### Author Response · Authors · 2025-06-02
>
> We thank reviewer `A835` for their thorough review. The reviewer highlights the important contributions provided by our pipeline and high quality dataset. We address the points raised by the reviewer below.
>
> ## Appendix
>
> The reviewer’s main concern has to do with the way the Appendix was used in our paper. While the way we structured the paper stemmed from an attempt to fit most of the content that we considered relevant in the 9 page limit, we fully agree with the reviewer that the current structure does not make for the best reading experience.
>
> We are committed to address this issue and will update our draft to improve the general flow and content imbalance between the main body of the paper and the Appendix, and in particular to make important results clearer in the main body of the paper.
>
> ## Under-explored sections
>
> The reviewer mentions that certain sections of the paper are under-explored. We believe this concern is related to the previous one, and will strive to make the different sections clearer. Specifically, for deduplication (Section 4.3, last paragraph, impact on data quality) we will mention that even the languages that seem to not benefit from deduplication see large performance uplifts during rehydration (that relies on deduplication); for filtering recipes (Section 4.4, statistics on each language on different corpora, different metrics etc.), we will include illustrative examples with some metrics across datasets; stopwords (Section 4.4.1, why at least two words from the stopwords list?) this follows from the original Gopher approach (we will clarify).
>
> ## Misc questions
>
> > The paper talks about “early-signal” benchmarks and “high-signal” benchmarks. Are they the same? Is there a difference?
>
> We consider there to be a difference: high-signal benchmarks allow us to properly compare different datasets/models with a certain degree of confidence regarding the meaningfulness of score differences. Early-signal benchmarks provide this “high-signal” behaviour in the early stages of pre-training, which is the setting we use for our experiments. That said, we do use these two terms somewhat interchangeably in the paper and will change this to be clearer in our draft.
>
> > Section 4.1: the mentioned “blocklist” was not questioned at all but simply accepted as fit for purpose?
>
> This blocklist is employed in previous literature, specifically in FineWeb (Penedo et al., 2024) and in RefinedWeb (Penedo et al., 2023). In the latter, a thorough analysis of this blocklist is provided. We will update our draft to point readers to RefinedWeb’s analysis.
>
> > Several times phrases like “the concurrent work HPLTv2.0” are mentioned. Why exactly is HPLTv2.0 singled out and labelled as “concurrent work”?
>
> This is a relevant work that was published at a late stage of the development of our own paper, which is why it was not included in, for example, our baseline datasets list that was used for task selection. We agree with the reviewer that always referring to it as “concurrent” can be excessive, and will update our draft accordingly.
>
> > The choice for GlotLID (paragraph 3 on page 5) needs a more comprehensive explanation.
>
> We will attempt to more clearly justify this decision in the main body of the paper. Our reasoning (as we detail in Appendix A.6.1) was primarily related to the larger number of labels supported (which allows us to have data in many more languages) as well as to some unique design decisions of GlotLID compared to other LID systems: it explicitly includes scripts, using script detection to narrow down possible labels and allowing us to tailor filtering to each script (in languages with multiple ones); UND labels for unsupported scripts (so that they are not assigned to another label); and specific “noise” labels such as text decoded with the wrong encoding, binary content, or misrendered PDFs, preventing it from being classified as a natural text language.
>
> > Section 4.4.3: I would’ve expected a discussion of mixed-languages documents here (or somewhere else in this paper).
>
> We believe mixed-language documents can be useful for multilingual training (they essentially provide “free” parallel data) and that language classifier confidence scores may also be affected by this type of data, but indeed we have not specifically discussed them in our work. Could the reviewer please clarify which specific aspects of mixed-language documents they would like to see discussed in the paper?
>
> ## Typography changes
>
> We deeply thank the reviewer for taking the time to compile different typography issues found in our work, which we will update with most of the proposed changes.

---

> > ### Comment · Reviewer_A835 · 2025-06-03
> > **Ack**
> >
> > Thanks for the comprehensive response that addresses my points.
> >
> > Regarding mixed-language documents: I was wondering if you attempt to detect them at all, if you do anything with them within the pipeline, if you perhaps throw them out if a certain threshold is reached etc. Here, by "mixed-language document", I don't only mean parallel or quasi-parallel data but also documents that primarily contain text in one language and, interspersed, there's text in other languages. This happens a lot or, at least several years ago, I've seen many of these documents. In other words, is assuming that the whole mixed-language document is, say, English (because this is the dominant language in the mixed-language document) appropriate or do documents of this class need special treatment?

---

### Decision · Program_Chairs · 2025-07-08

**Decision:**

Accept

**Comment:**

This is an impressive pre-training dataset of 1000 languages of size of 19TB. This is a very important contribution especially for numerous languages where pre-training dataset is scarce and hard to collect and clean-up from CommonCrawl.
The authors discuss numerous challenges they faced starting from tokenizers for languages with non-space speparated scripts, language recognition, to finding evaluation signals for low-resource languages.

All the reviewers and I appreciate this work and the only concerns were about format and what is highlighted in the main paper vs the massive appendix. Clearly a solid contribution for open LM science.

**This paper went through ethics reviewing. Please review the ethics decision and details below.**
Decision: Acceptance (if this paper is accepted) is conditioned on addressing the following in the camera-ready version
Details: The paper should include a potential risk statement.